# Byte Pair Encoding for Efficient Time Series Forecasting

**Leon Götz**[1 2] **Marcel Kollovieh**[2] **Stephan Günnemann**[2] **Leo Schwinn**[2 3]

## Abstract

Existing time series tokenization methods predominantly encode a constant number of samples into individual tokens. This inflexible approach can generate excessive tokens for even simple patterns like extended constant values, resulting in substantial computational overhead. Inspired by the success of byte pair encoding, we propose the first pattern-centric tokenization scheme for time series analysis. Based on a discrete vocabulary of frequent motifs, our method merges samples with underlying patterns into tokens, compressing time series adaptively. Exploiting our finite set of motifs and the continuous properties of time series, we further introduce conditional decoding as a lightweight yet powerful post-hoc optimization method, which requires no gradient computation and adds no computational overhead. On recent time series foundation models, our motif-based tokenization improves forecasting performance by $40\%$ and boosts efficiency by $2314\%$ on average. Conditional decoding further reduces MSE by up to $48\%$. In an extensive analysis, we demonstrate the adaptiveness of our tokenization to diverse temporal patterns, its generalization to unseen data, and its meaningful token representations capturing distinct time series properties, including statistical moments and trends.

## 1. Introduction

Transformer architectures have gained increasing relevance in time series processing, demonstrating impressive performance. Here, a key prerequisite for strong performance is effective tokenization – dividing the input into smaller units and embedding them in a high-dimensional space.

Yet, current tokenization schemes in time series processing exhibit considerable limitations: Early works embed each

[1]Volkswagen AG [2]Technical University of Munich [3]Helmholtz AI. Correspondence to: Leon Götz <leon.goetz@volkswagen.de>.

*Proceedings of the 43rd International Conference on Machine Learning*, Seoul, South Korea. PMLR 306, 2026. Copyright 2026 by the author(s).

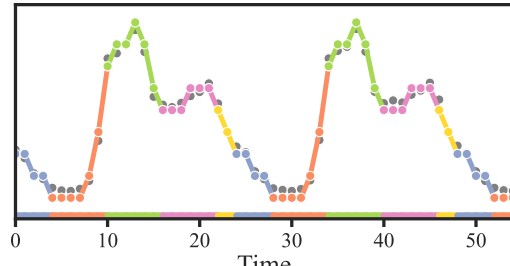

*Figure 1.* Motif-based tokenization transforms time series data (gray) through a two-step process: 1) quantizing samples into discrete bins, 2) merging recurring patterns of variable length into representative motifs (identical motifs share the same color). Motif repetition is highlighted by their x-axis projection.

individual time step as a token, creating a fundamentally inefficient representation, where every token captures little temporal information. This results in very long token sequences, imposing a substantial computational burden in the transformer architecture (Götz et al., 2025). Splitting the time series into fixed-length subsequences, called patches, mitigates both issues (Nie et al., 2023). However, rigid patches cannot adapt to diverse temporal patterns of varying lengths and complexities (Woo et al., 2024).

Inspired by adaptive pattern-based tokenization schemes in natural language processing (NLP) (Sennrich et al., 2016), we go beyond previous work and propose the first pattern-centric tokenization for time series, illustrated in figure 1. Our contribution is threefold:

**Adaptive tokenization for time series** We provide a novel tokenization strategy based on a discrete vocabulary of frequent time series motifs. Our method merges samples with underlying patterns into single tokens, enabling adaptive compression while maintaining a small upper-bounded discretization error. On the recently proposed Chronos foundation model, our tokenization improves forecasting performance by $40\%$ and boosts efficiency by $2314\%$ on average in a zero-shot setting.

**Conditional decoding** We introduce conditional decoding as a post-hoc optimization method to further improve forecasting performance by exploiting the continuous properties of time series to effectively remove the discretization error induced by our motifs. Conditional decoding is lightweight, requires no gradient computation, introduces no additional overhead during inference, and can be combined with any

pretrained time series model with a discrete output vocabulary. We demonstrate its effectiveness in large foundation models, increasing forecasting performance up to $48\%$.

**Empirical analysis** In an extensive empirical study, we demonstrate the zero-shot generalization capability of our tokenizer and its ability to automatically adapt to diverse temporal patterns and datasets. We link distinct time series characteristics, including statistical moments and trends, to our token representations and show that complex motifs benefit forecasting quality.

## 2. Related work

In recent years, transformer models have shown impressive performance in time series forecasting. While initial work focuses on efficient attention mechanisms and domain-specific architectures (Wu et al., 2021; Zhou et al., 2022), universal foundation models have been proposed lately (Garza & Mergenthaler-Canseco, 2023; Das et al., 2023; Rasul et al., 2023; Ansari et al., 2024; Woo et al., 2024; Goswami et al., 2024; Gao et al., 2024; Cohen et al., 2024; Liu et al., 2024a;b; 2025). These models are usually trained on billions of tokens and exhibit high zero-shot performance. However, all these transformer architectures rely on two basic tokenization techniques: using every sample as a token or extracting fixed-length patches from time series.

**Sample-based tokens** Most early works on transformer models for time series processing extract tokens for every time step, usually as a slice of a multivariate time series (Zhou et al., 2021; Wu et al., 2021; Zhou et al., 2022; Liu et al., 2022a;b; Cirstea et al., 2022). These tokens are linearly transformed into a continuous embedding space. Inspired by the success of discrete token embeddings in NLP, the recently proposed Chronos foundation model (Ansari et al., 2024) quantizes a univariate time series into bins and embeds them using learned vectors. This way, the authors transform forecasting from a regression task to classifying the next time step from a discrete vocabulary (Torgo & Gama, 1997). Masserano et al. (2025) use a wavelet-transformation-based approach for tokenization. Generating tokens for every time step has two major limitations: First, the large number of tokens imposes a substantial computational burden in transformers, especially for long sequence processing (Godahewa et al., 2021; Ansari et al., 2024). Second, every token captures only little information about temporal patterns (Chen et al., 2025).

**Patch-based tokens** Inspired by the success of patching in computer vision (Dosovitskiy et al., 2021), Nie et al. (2023) adapt this approach to time series, where multiple samples of a univariate time series are combined into individual tokens. Most subsequent works embed the patches into a continuous space using learned transformations (Zhang &

Yan, 2023; Nie et al., 2023; Wang et al., 2024; Wu et al., 2024; Das et al., 2023; Woo et al., 2024; Goswami et al., 2024; Gao et al., 2024; Cohen et al., 2024; Auer et al., 2025; Liu et al., 2024a;b; 2025). More advanced approaches learn a discrete codebook of patches (Talukder et al., 2024; Chen et al., 2024) using vector quantized variational autoencoder approaches (van den Oord et al., 2017). Patches generally compress the time series and capture local temporal information. However, due to their fixed length and stride, rigid patches can not adapt to varying temporal patterns in a sequence. This is particularly important for foundation models, as they aim to generalize to previously unseen data in zero-shot settings. To mitigate this, Woo et al. (2024) utilize different patch lengths for datasets sampled in different granularities, e.g., minutely or hourly. Their approach requires training of a new embedding transformation for every granularity. Shi et al. (2025) use multiple predefined patch lengths to predict time series in a mixture of experts approach and Wang et al. (2025) derive the patch length from the single dominant frequency in a time series. Even these approaches fail to capture diverse patterns of arbitrary lengths within a sequence (see section 5.4).

**Motif-based tokens** Motif-based tokenization utilizes a discrete vocabulary of recurring patterns. In NLP, byte pair encoding hierarchically extracts pairs of character-bytes to tokenize a sentence (Shibata et al., 1999; Sennrich et al., 2016). Elsner et al. (2024) extend this concept from 1d-sequences to tokenizing images. Moreover, tokenization based on discrete motifs has proven to be a good inductive bias for high-dimensional distribution learning as it reduces the combinatorial complexity (Sommer et al., 2023). Similarly, classical time series literature explored symbolization and pattern discovery techniques (Lin et al., 2003; Berndt & Clifford, 1994). Yet, data-dependent tokenization techniques as proposed in this work remain unexplored for machine-learning-based time series analysis.

## 3. An adaptive tokenization approach for time series

Despite recent advances in time series processing, current tokenization methods lack efficiency or fail to capture distinct temporal patterns within sequences (Ekambaram et al., 2024; Woo et al., 2024). We propose an efficient tokenization method using a vocabulary of frequent motifs as depicted in figure 2. Our algorithm combines samples with underlying patterns of varying complexity into single tokens. Its adaptive compression of time series enables efficient long sequence processing. We list pseudocode in appendix A.

Let $\mathcal{D} = \{z^i\}_{i=1}^N$ be a family indexed by $i = 1, \ldots, N$ of $N$ univariate real-valued time series $z = (z_1, \ldots, z_n) \in \mathbb{R}^n$ of length $n$. We normalize each series to have zero mean and unit standard deviation. A neural network $\mathbf{f}_{\boldsymbol{\theta}} : \mathbb{N}^{t_{\text{in}}} \rightarrow \mathbb{N}^{t_{\text{out}}}$

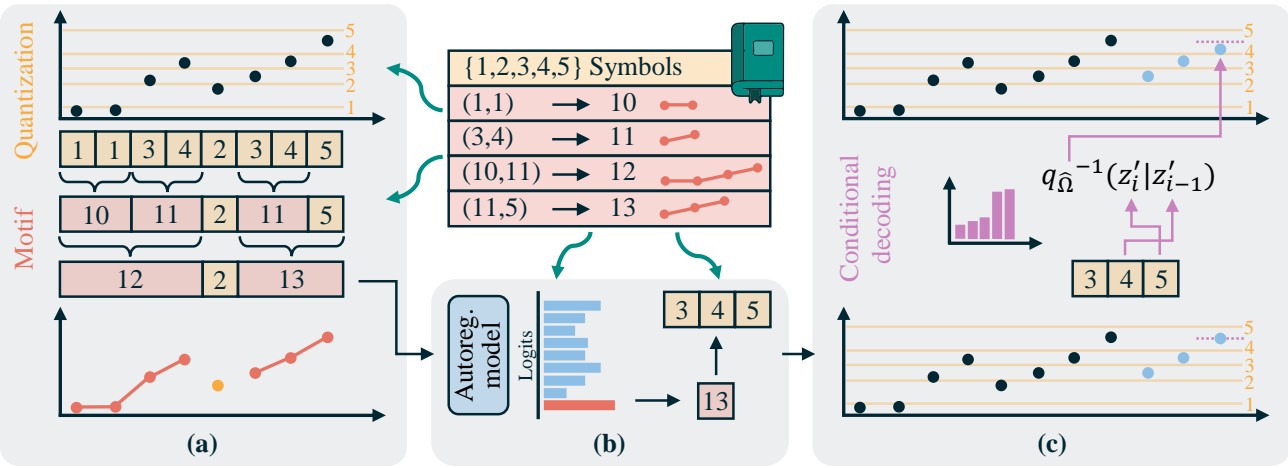

*Figure 2.* **(a)** Our motif-based tokenization first quantizes a time series into symbols and finds recurring motifs as tokens, building a discrete vocabulary. **(b)** Based on the compressed motif sequence, a neural network forecasts the time series through a categorical distribution over our vocabulary. **(c)** Finally, we propose conditional decoding to reduce the discretization error when transforming tokens back to their continuous representation.

with parameters $\boldsymbol{\theta}$ predicts $t_{\text{out}}$ token IDs from $t_{\text{in}}$ token IDs. Thereby, the tokens are generated by our tokenizer $\mathbf{g} : \mathbb{R}^n \to \mathbb{N}^t$ from a time series $z$.

Our tokenization consists of two steps:

$$\mathbf{g}(z) = \mathbf{m}_\Psi \circ \mathbf{q}_\Omega(z) \qquad (1)$$

where:

| | |
|---|---|
| $\mathbf{q}_\Omega : \mathbb{R}^n \to \mathbb{N}^n$ | quantizes the time series into a sequence of discrete symbols, |
| $\mathbf{m}_\Psi : \mathbb{N}^n \to \mathbb{N}^t$ | compresses the sequence based on a discrete vocabulary of temporal motifs, |
| $\Omega, \Psi$ | vocabulary of quantized symbols and motifs, respectively. |

### 3.1. Discretization of real-valued time series

Generalizing the approach from Ansari et al. (2024), we sample $M$ equiprobable discretization intervals $\Omega = \{C^{-1}\left(\frac{j}{M}\right)\}_{j=1}^{M}$, where $C^{-1}$ is the inverse cumulative distribution of the probability distribution $P$. In practice, we experiment with truncated uniform distributions in $[\omega_{\text{lb}}, \omega_{\text{ub}}]$, Gaussian distributions, and the precise data distribution $P(\mathcal{D})$ for binning. Utilizing the boundaries, we encode the time series $z$ into a sequence of discrete symbols:

$$\mathbf{q}_\Omega(z) = \{q_\Omega(z_i) \mid z_i \in z\},$$

$$\text{where} \quad q_\Omega(z_i) = \begin{cases} 1 & \text{if } z_i \leq \omega_1 \\ j & \text{if } \omega_{j-1} < z_i \leq \omega_j \end{cases}, \quad \omega_j \in \Omega\,. \qquad (2)$$

For decoding symbol IDs back to time series samples, we use $\hat{\Omega} = \{C^{-1}\left(\frac{j-0.5}{M}\right)\}_{j=1}^{M}$, where $\hat{\omega}_j \in \hat{\Omega}$ is the probabilistic center of $[\omega_{j-1}, \omega_j]$. Within the tokenization range, the quantization error can be upper bounded

as $\delta_{\max} = \max_{1<j\leq M} \max(\hat{\omega}_j - \omega_{j-1}, \omega_j - \hat{\omega}_j)$. For uniform binning, the probabilistic center is equal to the geometric center, and the maximum error simplifies to $\delta_{\max} = (\omega_{\text{ub}} - \omega_{\text{lb}})(2M)^{-1}$. Besides the $M$ token IDs representing quantized time series samples, we further introduce a masking token `MASK` to account for missing samples and an `EOS` token, which we insert at the end of time series.

### 3.2. Vocabulary of temporal motifs

Originally proposed for compressing raw byte sequences (Gage, 1994), byte pair encoding has been widely used in NLP to compress character sequences into subwords (Sennrich et al., 2016). Here, we generalize the byte pair compression algorithm to extract temporal patterns from our discretized time series. To this end, we iteratively build a vocabulary $\Psi$ of frequent time series motifs: Given a dataset $\mathcal{D}' = \{\mathbf{q}_\Omega(z^i) \mid z^i \in \mathcal{D}\}$ of quantized time series, we extract the most frequent adjacent token IDs $(z'_i, z'_{i+1})$, assigning a new token ID $z'_{\text{new}}$, which we add to our set of patterns $\Psi$:

$$\Psi^{(l+1)} \leftarrow \Psi^{(l)} \cup \{(z'_i, z'_{i+1}) \to z'_{\text{new}}\}\,. \qquad (3)$$

This process hierarchically finds distinct temporal motifs as discrete tokens and is locally optimal in every step. We build our vocabulary until the new tokens occur less frequently than $p_{\min}$ in $\mathcal{D}'$. This ensures that a minimum number of occurrences are available for a neural network to learn the motifs. Leveraging our vocabulary, we compress a quantized time series into a sequence of motifs:

$$\mathbf{m}_\Psi(z') = \{\psi(z') \mid \psi \in \Psi\}, \quad \mathbf{m}_\Psi : \mathbb{N}^n \to \mathbb{N}^t\,. \qquad (4)$$

The compression is highly flexible as motifs of different lengths and complexities are mapped to single tokens. We

define the average compression at the sequence level as $\bar{c} = n/t$. Our algorithm has linear complexity in sequence length $O(|\Psi| \cdot n)$, enabling long sequence processing.

### 3.3. Conditional decoding

We propose our novel conditional decoding to universally improve the forecasting quality of models with discrete output vocabularies. To decode a token sequence, such as the predictions of a model, we invert the tokenization $\mathbf{g}$. In this process when inverting $\mathbf{q}_\Omega$, we previously leveraged the bin centers $\hat{\omega}_j \in \hat{\Omega}$ to transform a quantized sequence $z'$ back to a time series $\hat{z}$. We introduce conditional decoding to reduce the overall quantization error. Specifically, we decode quantized time series samples $z'_i$ conditioned on the previous sample $\hat{z}_i = q_{\hat{\Omega}}^{-1}(z'_i \mid z'_{i-1})$. To this end, we set parameters $\hat{\Omega} = \{\hat{\omega}_{j,k} \mid j, k \in \{1, \ldots, M\}\}$, where $q_{\hat{\Omega}}^{-1}(z'_i = j \mid z'_{i-1} = k) = \hat{\omega}_{j,k}$ to minimize $\|z_i - \hat{z}_i\|_2^2$:

$$\min_{\hat{\Omega}} \sum_{(z,z') \in \mathbf{D}} \sum_{i=2}^n \left\| z_i - q_{\hat{\Omega}}^{-1}(z'_i \mid z'_{i-1}) \right\|_2^2, \tag{5}$$

$$\text{where} \quad \mathbf{D} = \{(\mathcal{D}_i, \mathcal{D}'_i) \mid i \in \{1, \ldots, N\}\}$$

consists of corresponding real-valued and quantized time series. Thereby, a single parameter $\hat{\omega}_{j,k}$ is given by the mean of the underlying time series samples $\tilde{z}$, minimizing the squared error:

$$\hat{\omega}_{j,k} = \frac{1}{|\mathcal{D}_{j,k}|} \sum_{\tilde{z} \in \mathcal{D}_{j,k}} \tilde{z}, \tag{6}$$

$$\text{where} \quad \mathcal{D}_{j,k} = \{z_i \mid (z, z') \in \mathbf{D}, z'_i = j, z'_{i-1} = k\}.$$

Intuitively, we adopt a unigram model to exploit the unique properties of our tokenization: the finite set of discrete symbols and the underlying continuous time series samples. Conditional decoding is lightweight and requires no gradient computation as we solve analytically for the global optimum. Further, it adds no additional inference cost[1] and is very small in practice with only $M^2$ parameters $\hat{\omega}_{j,k} \in \hat{\Omega}$ relying on the first-order Markov assumption[2]. Conditional decoding can be combined with any pretrained time series model with a discrete output vocabulary and considerably improves forecasting performance in our experiments.

### 3.4. Model architecture

As we represent continuous time series as a sequence of discrete motifs, we can rely on recent advances in transformer architectures in natural language processing. These architectures transform token IDs from our discrete vocabulary $\mathcal{V} = \Omega \cup \Psi \cup \{\texttt{MASK}, \texttt{EOS}\}$ into $d$-dimensional space using learned embedding tables $E \in \mathbb{R}^{|\mathcal{V}| \times d}$. We optimize

---

[1] It uses the same $O(1)$ dictionary lookup as normal decoding.

[2] We analyze higher-order models in appendix C.3.

the parameters $\boldsymbol{\theta} \in \boldsymbol{\Theta}$ of our model $\mathbf{f}_{\boldsymbol{\theta}}$ on autoregressive next token prediction of our tokenized sequence $z'' = \mathbf{g}(z)$. Our model thereby predicts a categorical distribution $p(z''_{t_{\text{in}}+1} \mid z''_{1:t_{\text{in}}})$ over our finite vocabulary of time series motifs $\mathcal{V}$. We impose a cross-entropy loss for distribution learning. To this end, we transform the regression task to a classification (Torgo & Gama, 1997). The discrete set of possible motifs reduces the combinatorial complexity and has proven to be a good inductive bias for distribution learning in the bio-medical domain (Sommer et al., 2023). In contrast to prior work (Ansari et al., 2024), our tokenizer enhances the efficiency as both model input and generated tokens are compressed time series representations. Models can utilize longer contexts while requiring fewer autoregressive iterations for a given prediction horizon. This is especially important for large foundation models and long sequence processing, imposing substantial computational demands.

## 4. Experiments

We systematically train different tokenizers and foundation models and evaluate them on 7 time series datasets in a zero-shot setting, demonstrating advantages of our motif-based representation over tokenizing every sample or utilizing patches. In appendix B, we provide further details.

**Datasets** For training our models and tokenizers, we utilize the recently proposed Chronos dataset (Ansari et al., 2024). It contains $11\,\text{M}$ time series with over $11\,\text{B}$ samples. Due to its diverse nature and size, this dataset is well-suited for training foundation models. We base our zero-shot evaluation on 7 commonly used time series datasets: ETTh1, ETTm1, Weather, Electricity, Traffic, Solar, and Fev-bench (Godahewa et al., 2021; Shchur et al., 2025). Note that Fev-bench is a combination of multiple datasets. We specify the exact configuration in appendix B.

**Tokenizers** We leverage 3 tokenizers with different numbers of quantization bins $M$. Further, we utilize a truncated uniform distribution from $\omega_{\text{lb}} = -5$ to $\omega_{\text{ub}} = 5$ for binning, spanning a range of 5 standard deviations. As a result, our tokenizers in table 1 feature different compression ratios, vocabulary sizes, and discretization errors. We build their vocabulary $\Psi$ on the same $100\,000$ randomly selected time series from the Chronos dataset with a total of $100\,\text{M}$ samples. To allow the model to learn all tokens, we constrain the motifs to occur at least $p_{\text{min}} = 1000$ times in the compressed data. In sections 5.5 and 5.6 and appendix C.1, we systematically ablate these choices.

*Table 1.* Tokenizers on the Chronos dataset with different quantization bins, vocabulary size, discretization error, and compression.

| Compression | $M$ | $|\mathcal{V}|$ | $\delta_{\text{max}}$ | $\bar{c}$ |
|---|---|---|---|---|
| low | 126 | 2445 | 0.040 | 2.08 |
| medium | 37 | 1675 | 0.135 | 3.18 |
| high | 22 | 1373 | 0.227 | 4.06 |

**Models** In our experiments, we explore our tokenization approach in foundation models operating in a zero-shot setting. We compare our motif-based tokenization with sample-based tokens in Chronos models (Ansari et al., 2024). Additionally, we implement a patch-based version of Chronos, where we alter only the tokenization method and replace the cross-entropy loss function with MSE, which is generally used for continuous patches. We select non-overlapping patches of length 4 with similar compression as our high-compression tokenizer and length 8, as recommended in recent literature (Goswami et al., 2024). Following these baseline models, we use the T5 architecture (Raffel et al., 2020) as backbone for our motif-based tokenization and train all models with the same number of tokens, gradient steps, and training settings. This way, we compare our motif-based tokenization to sample-based tokenization and patches in an isolated setting, ensuring that the tokenization method is the only difference between architectures. Following Chronos models, we propose models with our tokenizer in 5 sizes ranging from tiny (8 M parameters) to large (710 M parameters). We evaluate on forecasting 64 time series samples, following Ansari et al. (2024). As context, we utilize 128 tokens for our and Chronos models and an equivalent input length of 384 time series samples for patch-based models. As literature references, we use patch-based MOMENT (Goswami et al., 2024), Moirai (Woo et al., 2024), Time-MoE (Shi et al., 2025), and LightGTS (Wang et al., 2025) foundation models. We restrict evaluation to models with available data and code for reproducibility.

## 5. Results

We first demonstrate improvements in forecasting performance and efficiency of our motif-based tokenization over existing methods. Next, we explore the adaptiveness of our tokenizer to diverse temporal patterns of different lengths and complexities and its generalization to unseen data. Finally, we link distinct time series properties, including statistical moments, to our token space. In appendix C.9, we visualize the learned motifs.

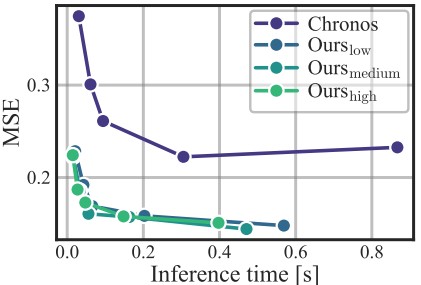

*Figure 3.* Zero-shot comparison between our motif-based and sample-wise tokenization (Chronos) on Electricity.

### 5.1. Efficiency improvements of adaptive tokenization

Chronos foundation models tokenize every sample of a time series, resulting in many tokens with little temporal information. Especially for large models, this induces substantial computational requirements. We compare our motif-based tokenization with Chronos models in 5 sizes from tiny to large using 3 tokenizers (see experimental settings in section 4). Chronos and our models are based on the same architecture, training strategy, and dataset. They only differ in tokenization.

In our zero-shot evaluation, our motif-based tokenization finds Pareto optimal points on all 7 datasets. We show in figure 3 that our tokenizer outperforms Chronos models with sample-based tokenization in forecasting quality and efficiency at the same time. We report our results in table 2, choosing the best Chronos model as reference. Among our 3 tokenizers and 5 model sizes, we illustrate two cases: 1) Selecting the best MSE, 2) Selecting the fastest model that is still better than the Chronos reference. Motif-based tokenization without conditional decoding improves MSE by 39.8 % and accelerates models 23.14 × on average. With conditional decoding, the improvements are even more substantial, with forecasting quality increasing by 48.0 % and model acceleration reaching 28.22 ×. On the Traffic dataset, Chronos models diverge during zero-shot testing, while our tokenizer still performs well, highlighting the generalization capability of motifs. We show full results in appendix C.2

*Table 2.* Motif-based tokenization with conditional decoding (cd) and without improves forecasting quality and accelerates models during zero-shot forecasting. We aim for two extremes: best MSE and fastest acceleration. Among Chronos models, we choose the best as reference. As our tokenization improves MSE while speeding up the model, we are able to choose small models while surpassing the forecasting quality of larger ones. **Best** in bold.

| Dataset | Chronos | Ours | | Ours$^{cd}$ | |
| --- | --- | --- | --- | --- | --- |
| | MSE | MSE$^{best}$ | Accel.$^{fastest}$ | MSE$^{best}$ | Accel.$^{fastest}$ |
| ETTh1 | 0.717 | 0.517 | 24.88× | **0.459** | **55.74×** |
| ETTm1 | 1.004 | 0.637 | **6.49×** | **0.449** | **6.49×** |
| Weather | 0.265 | 0.251 | 0.26× | **0.236** | **3.58×** |
| Electricity | 0.222 | 0.150 | **11.20×** | **0.144** | **11.20×** |
| Traffic | 2.717 | 0.591 | **56.66×** | **0.574** | **56.66×** |
| Solar | 1.270 | 0.439 | **4.69×** | **0.371** | **4.69×** |
| Fev-bench | 1.489 | 1.006 | **57.80×** | **0.756** | **57.80×** |

*Table 3.* Benchmarking our motif-based tokenization with conditional decoding (cd) and without against our patch-based Chronos baseline, MOMENT, Moirai, Time-MoE, and LightGTS models, based on zero-shot forecasting quality (MSE). In line with table 2 we report the best among our tokenizers. We highlight values that are  worse  than our method.

| Dataset | Ours | Ours$^{cd}$ | Chronos$_{patch}^{len=4}$ | | | | | Chronos$_{patch}^{len=8}$ | | | | | MOMENT | | | Moirai | | | Time-MoE | | LightGTS |
|---|---|---|---|---|---|---|---|---|---|---|---|---|---|---|---|---|---|---|---|---|---|
| | | | tiny | mini | small | base | large | tiny | mini | small | base | large | small | base | large | small | base | large | base | large | |
| ETTh1 | 0.517 | 0.459 | 0.525 | 0.474 | 0.453 | 0.470 | 0.384 | 0.426 | 0.420 | 0.446 | 0.400 | 0.379 | 0.765 | 0.732 | 0.693 | 0.465 | 0.396 | 0.397 | 0.767 | 0.789 | 0.391 |
| ETTm1 | 0.637 | 0.449 | 0.879 | 0.912 | 0.916 | 1.099 | 0.666 | 0.800 | 0.647 | 0.906 | 0.704 | 0.608 | 0.700 | 0.710 | 0.665 | 0.710 | 0.600 | 0.548 | 0.714 | 0.733 | 0.813 |
| Weather | 0.251 | 0.236 | 0.425 | 0.319 | 0.356 | 0.601 | 0.374 | 0.273 | 0.305 | 0.264 | 0.278 | 0.284 | 0.275 | 0.249 | 0.240 | 0.193 | 0.161 | 0.245 | 0.286 | 0.298 | 0.157 |
| Electricity | 0.150 | 0.144 | 0.249 | 0.250 | 0.227 | 0.214 | 0.162 | 0.220 | 0.170 | 0.203 | 0.169 | 0.146 | 0.887 | 0.888 | 0.852 | 0.212 | 0.163 | 0.146 | 0.942 | 0.944 | 0.230 |
| Traffic | 0.591 | 0.574 | 0.766 | 0.808 | 0.762 | 0.756 | 0.624 | 0.731 | 0.645 | 0.685 | 0.680 | 0.625 | 1.458 | 1.534 | 1.386 | 0.645 | 0.406 | 0.427 | 1.571 | 1.556 | 0.618 |
| Solar | 0.439 | 0.371 | 1.353 | 1.047 | 1.004 | 1.258 | 0.682 | 0.856 | 0.870 | 0.926 | 0.841 | 0.556 | 0.837 | 0.832 | 0.820 | 1.019 | 0.947 | 1.108 | 1.063 | 0.959 | 0.611 |
| Fev-bench | 1.006 | 0.756 | 0.869 | 0.924 | 0.893 | 0.952 | 0.788 | 0.784 | 0.720 | 0.817 | 0.757 | 0.680 | 1.282 | 1.268 | 1.240 | 0.886 | 0.831 | 0.785 | 1.344 | 1.341 | 0.918 |

and further compare our zero-shot motif-based models with state-of-the-art models that are directly trained on the respective datasets. Remarkably, our approach generates the best forecasts in 19 out of 25 cases without fine-tuning.

## 5.2. Comparison with patch-based methods

**Patch-based Chronos** Patching, which involves extracting fixed-length subsequences as tokens, compresses the time series and captures local temporal information (Nie et al., 2023). However, patches are rigid and non-adaptive to diverse time series patterns. Here, we compare our adaptive motif-based tokenization with our patch-based Chronos baseline in an isolated setting, where tokenization is the only difference between models. Except for ETTh1 and Fev-bench, our tokenization method outperforms all patch-based Chronos models in our isolated comparison in table 3. Motif-based tokenization with conditional decoding increases forecasting quality by 12.7 % on average across all datasets. These results highlight the potential of byte pair encoding for time series.

**Beyond Chronos** We further compare with patch-based literature foundation models MOMENT, Moirai, Time-MoE, and LightGTS in a zero-shot setting. Our tokenizer consistently outperforms all MOMENT and Time-MoE models, and generates better forecasts than LightGTS and Moirai on 5 out of 7 and 4 out of 7 datasets, respectively. Given that these models utilize different backbones and datasets, their performance lead in the remaining cases may stem from these systemic advantages rather than the tokenization scheme itself. Nevertheless, these results demonstrate that our tokenizer enables performance competitive with state-of-the-art foundation models. In section 5.4, we explore the compression of our motif-based tokenization.

## 5.3. Conditional decoding

Recently emerging foundation models show impressive performance but are expensive to train (Ansari et al., 2024). We propose conditional decoding as a lightweight yet powerful post-hoc optimization method to enhance a model's forecasting quality. Conditional decoding adds no computational overhead during inference and does not require gradient computation for training. Instead, we analytically compute the global optimum for its few parameters according to

equation (6). For our experiments, we utilize 3 tokenizers with different compression (see table 1), 7 datasets, and models in size small. In the following, we train conditional decoding to dequantize the models' forecasts on the respective train set and evaluate on the test set. Here, data- and model-dependent conditional decoding may act as an effective domain adaptation method. Later in appendix C.3, we evaluate data- and model-independent conditional decoding in pure zero-shot settings.

Conditional decoding consistently improves forecasting quality in figure 4 in all of our experiments. On the ETTm1 dataset and our tokenizer with high compression, conditional decoding reduces MSE by 44.3 % with only 484 trainable parameters. In appendix C.3, we provide additional results and further investigate conditional decoding in a data- and model-independent setting. There, conditional decoding mitigates on average 31.9 % and up to 96.9 % of our tokenizer's quantization error, enabling us to build tokenizers with even higher compression.

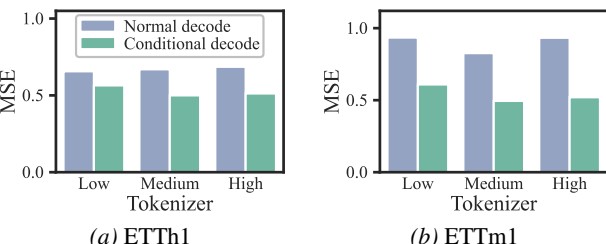

*(a)* ETTh1      *(b)* ETTm1

*Figure 4.* Conditional decoding improves forecasting quality for 3 tokenizers in small models on 2 datasets.

## 5.4. Adaptive compression of diverse time series

Here, we analyze the efficiency benefits of adaptive tokenization in detail. Temporal patterns differ in length and complexity among datasets and within time series (Ekambaram et al., 2024; Woo et al., 2024). While rigid patches are generally unable to capture these inter- and intra-series variations by employing fixed compression rates, our motif-based tokenization natively exploits these diverse patterns, compressing them adaptively. Here,

*Table 4.* Average compression of our medium tokenizer on 5 datasets.

| Dataset | $\bar{c}$ |
|---|---|
| ETTh1 | 3.48 |
| ETTm1 | 4.59 |
| Weather | 23.15 |
| Electricity | 3.95 |
| Traffic | 3.30 |

we analyze our medium compression tokenizer (see table 1). The Weather dataset contains patterns of various complexities, which we illustrate in figures 5b to 5d. Here, our tokenizer compresses motifs of different lengths into single tokens, achieving compressions from 8.13 up to 22.26. Less complex patterns result in higher compression, while more complex patterns are tokenized more fine-grained. In figure 11, we demonstrate this adaptive intra-series compression on 4 other datasets. Among datasets, our tokenizer reaches average compressions of 3.30 on Traffic and 23.15 on Weather in table 4. Further, ETTh1 and ETTm1 are sampled with different frequencies but from the same process. The higher compression on ETTm1 indicates that our tokenizer is agnostic to the sampling frequency. All these results highlight the flexibility of our motif-based tokenization. Compared to MOMENT with patch length 8 and 7 other patch-based models in table 19, our tokenizer can yield substantially greater compression by adapting to time series structure. In appendix C.4, we further investigate relations between compression of input data and generated tokens and find linear dependencies. We also showcase even higher compressions up to 128. Note that we report efficiency gains in inference time for our main experiments. Here, we investigate adaptive compression of motif-based tokenization at time series level, which directly translates to real-world speed up by requiring fewer autoregressive generation steps. Tokenization overhead is negligible with $< 0.5\,\%$ in runtime of our fastest models.

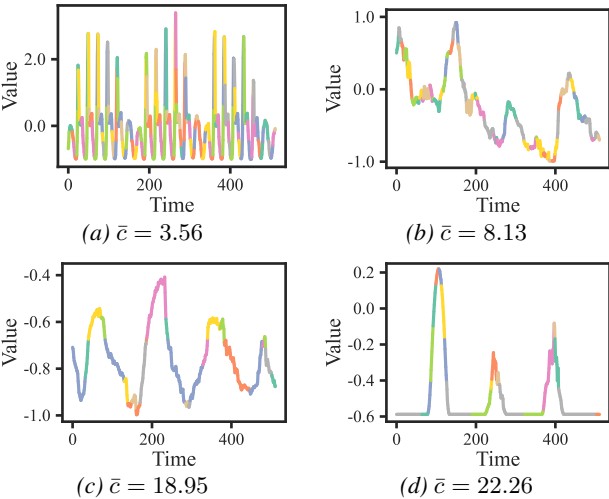

*Figure 5.* Our adaptive tokenizer **(a)** exploits periodically recurring motifs on the Traffic dataset and **(b-d)** compresses time series adaptively depending on pattern complexity on the Weather dataset.

## 5.5. Vocabulary complexity and generalization

Longer motifs benefit the compression and efficiency of our tokenizer. Here, we systematically explore factors influencing the vocabulary complexity and generalization ability. We show that longer motifs are more expressive and enhance

forecasting quality. We list further insights in appendix C.5. In appendix C.6, we investigate robustness to noise, extreme values, and generalization to non-stationary time series.

**Quantization granularity** A lower number of quantization bins $M$ reduces the complexity of the time series, resulting in longer motifs and a smaller vocabulary (see table 1). However, fewer quantization bins also increase the quantization error, potentially failing to capture important nuances and compromising forecasting quality. In table 12 and figure 9, we utilize 3 tokenizers with different quantization granularities without conditional decoding, across 5 model sizes, and 7 datasets to analyze this tradeoff.

In 25 out of 35 settings, our tokenizer with high compression and the largest quantization error leads to the best MSE. This experiment indicates that longer, more expressive motifs benefit forecasting, despite higher quantization error. Moreover, as shown in section 5.3, the quantization error can be largely removed with conditional decoding.

**Token occurrence** There is an inherent tradeoff in tokenization: longer, more complex motifs (created by a high number of recursive merges) naturally occur less frequently in the training data. In the limit, the whole dataset can be represented by a single motif. While setting a lower minimum occurrence threshold $p_{min}$ allows the vocabulary to capture more complex patterns, these rarer motifs may provide insufficient learning examples for the model to reliably recognize them. Here, we vary $p_{min}$ from 1000 to 128 000 training 8 different tokenizers. These tokenizers feature different vocabulary complexity and compression, as in tables 5 and 20, but have the same quantization error. We base our variations on our medium tokenizer and utilize small models. Our results on Electricity and Traffic in figure 6 indicate an optimal tradeoff. A minimum motif occurrence of $p_{min} = 4000$ times among 100 M time series samples represents a good balance. Generally, more complex motifs with higher compression result in the best MSE.

*Table 5.* Tokenizers on the Chronos dataset with different token occurrence, vocabulary size, and compression.

| $p_{min}$ | $|\mathcal{V}|$ | $\bar{c}$ |
|---|---|---|
| 1000 | 1675 | 3.18 |
| 8000 | 373 | 2.50 |
| 32 000 | 158 | 2.08 |
| 128 000 | 78 | 1.66 |

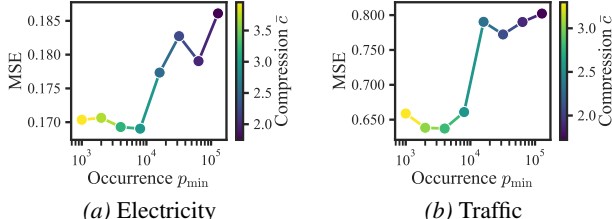

*(a)* Electricity     *(b)* Traffic

*Figure 6.* Varying token occurrence $p_{min}$ influences forecasting quality for small models. More complex motifs improve MSE.

**Token level analysis** Here, we demonstrate on the token level that complex motifs are a better representation

for time series generation than their simpler counterparts. To this end, we correlate motif length with token-wise MSE of time series forecasts. We utilize our medium compression tokenizer in a small model. On all 5 datasets, we observe negative correlation coefficients $\rho$ in table 6. Therefore, the generation of longer, more expressive motifs enhances forecasting quality. These results align with our previous findings.

*Table 6.* Correlation $\rho$ of token-level compression and MSE.

| Dataset | $\rho$ |
|---|---|
| ETTh1 | $-0.26$ |
| ETTm1 | $-0.24$ |
| Weather | $-0.26$ |
| Electricity | $-0.07$ |
| Traffic | $-0.27$ |

### 5.6. Training dataset size

Here, we explore how much data is required to train an efficient motif-based tokenizer. In general, larger datasets better approximate the true distribution of patterns, resulting in more complete vocabularies of motifs $\Psi$. To this end, we train our 3 tokenizers on Chronos dataset subsets ranging from 1000 to $1\,\text{M}$ time series and scale $p_{\min}$ accordingly. Increasing the dataset size improves forecasting quality, as our results in table 7 show (averaged across 3 tokenizers on 5 evaluation datasets). As expected, motifs extracted from a larger sample size are less noisy and generalize better. This is also evident in the decreasing vocabulary size and compression, indicating a smaller, more universal set of motifs. With $1\,\text{M}$ time series, our tokenizer is highly sample-efficient, requiring less than $10\,\%$ of Chronos data for vocabulary generation. We show full results and similar findings for conditional decoding in appendix C.7.

*Table 7.* Influence of training dataset size $N$ on tokenizer vocabulary size, compression, and forecasting quality.

| $N$ | $|\mathcal{V}|$ | $\bar{c}$ | MSE |
|---|---|---|---|
| $1\,\text{k}$ | 2127 | 3.16 | 0.569 |
| $10\,\text{k}$ | 1853 | 3.10 | 0.560 |
| $100\,\text{k}$ | 1831 | 3.11 | 0.555 |
| $1\,\text{M}$ | 1827 | 3.11 | 0.533 |

### 5.7. Learned token representations

Time series have distinct properties such as periodicity, offsets, and trends. A meaningful token representation should model these characteristics. Our motif-based tokenization

captures periodicity by design, mapping similar patterns at different positions in a time series to the same token. This is qualitatively shown in figures 1 and 5a. Moreover, we analyze the token embedding space $E$ by doing a principal component analysis in figures 7 and 20. The learned embeddings successfully capture the values of quantized symbols in $\Omega$ (a), which are separated from MASK and EOS tokens. The embedding space further models the mean (b) and standard deviation (c) of motifs in $\Psi$ in orthogonal dimensions, indicating a good separation of these properties. For motifs with high standard deviation, the model distinguishes between linear and quadratic trends. Finally, motif length is implicitly learned and modeled in the same dimension as the standard deviation, as constant patterns with low standard deviation are likely longer.

Our method builds time series motifs hierarchically, where each child token is formed from two parents. Intuitively, a child should be close to its first parent in the embedding space, since the model can predict either the child directly or its parents as a sequence. The average cosine similarity across all tokens is $0.072$, while parent–child pairs show a much higher value of $0.475$. We illustrate these relations in figure 21, where parents and children are shifted along the motif length axis.

In summary, our results confirm that our motif vocabulary yields meaningful time series representations.

## 6. Conclusion

In this work, we propose the first pattern-centric tokenization for the time series domain. Our method leverages recurring discrete motifs as tokens and improves forecasting quality and efficiency over existing methods. We further introduce conditional decoding as a lightweight, domain-specific post-hoc optimization method and show its performance gains in large foundation models. We demonstrate our tokenizer's adaptability to patterns of different complexities and show that the learned token embeddings capture meaningful representations of time series properties, including statistical moments and trends. Finally, our thorough investigation reveals key tradeoffs balancing tokenizer

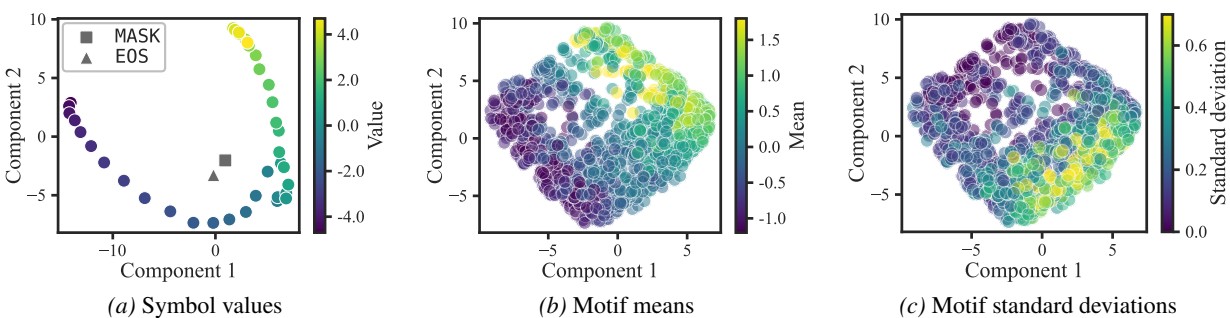

*(a)* Symbol values      *(b)* Motif means      *(c)* Motif standard deviations

*Figure 7.* Principal component analysis of token embeddings of our medium tokenizer in a small model.

complexity and generalization: discretization granularity presents a dual effect on compression - fewer bins increase discretization error but also make patterns more frequent, potentially improving both learnability and compression; training data size influences how well the discovered motifs generalize, with smaller datasets being insufficient to learn robust representations of rare motifs. However, with sufficient data, longer and more complex motifs can significantly reduce prediction error, ultimately enhancing compression efficiency. We hope our motif-based tokenization will have a positive effect on reducing the resource consumption and environmental impact of time series models.

**Limitations**  In our work, we do not conduct hyperparameter search for T5 models due to the high computational cost of training large foundation models. We expect even better results with optimized settings. Moreover, future work can utilize more recent transformer architectures.

## Impact statement

Motif-based tokenization and conditional decoding demonstrate large accelerations and considerable quality gains throughout a broad range of experiments. We hope that our novel tokenization scheme will contribute to better and more sustainable time series models.

## Disclaimer

The results, opinions, and conclusions expressed in this publication are not necessarily those of Volkswagen Aktiengesellschaft.

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

# A. An adaptive tokenization approach for time series

We provide pseudocode for generating a vocabulary of motifs and utilizing the motifs to tokenize a time series.

---

**Algorithm 1** Motif vocabulary generation according to equation (3).

---

**Input:** Dataset of discretized time series $\mathcal{D}'$, minimum motif occurrence $p_{\min}$
**Output:** Motif vocabulary $\Psi$

  $\Psi \leftarrow \{\}$                                                        $\triangleright$ Initialize empty vocabulary
  $z'_{\text{new}} \leftarrow M + 2$            $\triangleright$ Account for quantized symbols and $\{\texttt{MASK}, \texttt{EOS}\}$
  **while** true **do**                                   $\triangleright$ Iteratively find motifs
    $\text{pair}, \text{cnt} \leftarrow \text{count}\,(z'_i, z'_{i+1})$ in $\mathcal{D}'$     $\triangleright$ Most frequent adjacent token pair and its count
    **if** $\text{cnt} \geq p_{\min}$ **then**
      $z'_{\text{new}} \leftarrow z'_{\text{new}} + 1$                          $\triangleright$ Allocate new token ID
      $\Psi[\text{pair}] \leftarrow z'_{\text{new}}$           $\triangleright$ Add new token to vocabulary $(z'_i, z'_{i+1}) \rightarrow z'_{\text{new}}$
      $\mathcal{D}' \leftarrow \mathcal{D}' \setminus \{\text{pair}\} \cup \{z'_{\text{new}}\}$           $\triangleright$ Replace new token in dataset $\mathcal{D}'$
    **else**
      **return** $\Psi$                              $\triangleright$ Token occurs to infrequent
    **end if**
  **end while**

---

**Algorithm 2** Tokenization of a discretized time series according to equation (4).

---

**Input:** Discretized time series $z'$, motif vocabulary $\Psi$
**Output:** Tokenized time series $z''$

  **for** $\psi$ **in** $\Psi$ **do**                       $\triangleright$ Iterate over motifs in order of vocabulary creation
    $\psi_{\text{key}}, \psi_{\text{value}} \leftarrow \psi$            $\triangleright$ $\psi$ made of key value mappings $(z'_i, z'_{i+1}) \rightarrow z'_{\text{new}}$
    **for** $(z'_i, z'_{i+1})$ **in** $z'$ **do**               $\triangleright$ Iterate over adjacent tokens in $z'$
      **if** $(z'_i, z'_{i+1})$ matches $\psi_{\text{key}}$ **then**           $\triangleright$ Adjacent tokens match motif
        replace $(z'_i, z'_{i+1})$ with $\psi_{\text{value}}$ in $z'$       $\triangleright$ Replace tokens with motif: shortens $z'$ by 1
      **end if**
    **end for**
  **end for**
  $z'' \leftarrow z'$
  **return** $z''$

---

# B. Experiments

In this section, we list additional information about our experimental settings and resources.

**Datasets** We train our models and tokenizers on the recently proposed Chronos dataset (Ansari et al., 2024). It contains 11 M time series with over 11 B samples. Time series are curated from 28 real-world datasets or are generated synthetically. Due to its diverse nature and size, this dataset is well-suited for training foundation models.

We base our zero-shot evaluation on 6 commonly used time series datasets and the Fev-bench benchmark covering different forecasting applications: *ETTh1* and *ETTm1* measure the power load and temperature of electric transformers in hourly and quarter-hourly granularity (Zhou et al., 2021). *Weather* consists of meteorological quantities such as air temperature and is recorded every 10 minutes in 2020.[3] *Electricity* measures the energy demand of households in hourly granularity (Godahewa et al., 2021). *Traffic* consists of hourly road occupancies in the San Francisco Bay Area (Godahewa et al., 2021). *Solar* measures the power production of photovoltaic plants in 10 minute intervals (Godahewa et al., 2021). *Fev-bench* is a broad time series forecasting benchmark composed of datasets from diverse domains (Shchur et al., 2025). We exclude sequences with missing values and those shorter than our input and prediction horizons. In total, we evaluate on 37 datasets from Fev-bench spanning 5 domains, as in table 8, and report averaged results.

*Table 8.* Datasets from the Fev-bench benchmark used in our experiments.

| Domain | Dataset |
|---|---|
| Cloud | BizITObs L2C 5T, BizITObs L2C 1H, BOOMLET 619, BOOMLET 772, BOOMLET 963, BOOMLET 1209, BOOMLET 1225, BOOMLET 1230, BOOMLET 1282, BOOMLET 1487, BOOMLET 1631, BOOMLET 1676, BOOMLET 1855, BOOMLET 2187, Redset 5T, Redset 1H |
| Economy | US Consumption 1M |
| Energy | ENTSO-e Load 15T, ENTSO-e Load 30T, ENTSO-e Load 1H, EPF-BE, EPF-DE, EPF-FR, EPF-NP, EPF-PJM, ERCOT 1D, ERCOT 1W, ETT 1H, GFC12, GFC14, GFC17, Solar with Weather 1H |
| Mobility | Loop Seattle 5t, Loop Seattle 1H, M-DENSE 1H, SZ Taxi 15T |
| Retail | M5 1D |

**Hyperparameters** For training our T5 models, we utilize the hyperparameters of Chronos (Ansari et al., 2024), which we list in table 9. We expect even better results of our tokenizer when performing hyperparameter tuning. However, this is very expensive for large foundation models.

*Table 9.* Hyperparameters of our T5 backbone models in 5 sizes from tiny to large.

| Hyperparameter | tiny | mini | small | base | large |
|---|---|---|---|---|---|
| **T5 models** | | | | | |
| Token dimension $d$ | 256 | 384 | 512 | 768 | 1024 |
| Encoder layers | 4 | 4 | 6 | 12 | 24 |
| Decoder layers | 4 | 4 | 6 | 12 | 24 |
| Heads | 4 | 8 | 8 | 12 | 16 |
| **Training** | | | | | |
| Seed | | | 2024 | | |
| Activation | | | ReLU | | |
| Dropout rate | | | 0.1 | | |
| Learning rate | | | 0.001 | | |
| Learning rate decay | | | linear | | |
| Gradient steps | | | 200 000 | | |
| Batch size | | | 256 | | |
| Optimizer | | | Adam (Kingma & Ba, 2015) | | |

---

[3] https://www.bgc-jena.mpg.de/wetter/

**Reproducibility of measurements**   In our zero-shot evaluations, we use the same data splits as Wu et al. (2021). We evaluate once and report results on the test set.

Regarding **predictive quality**, we report MSE standard deviations for our most common experimental settings here, as repeating all experiments multiple times is computationally infeasible for large foundation models. To this end, we train small models with 6 randomized seeds with our medium compression tokenizer and Chronos baselines with sample-based tokenization and patches of length 8. Our zero-shot evaluation on 5 datasets in table 10 demonstrates low average MSE standard deviations of $6.7\,\%$ for our motif-based tokenization, $4.2\,\%$ for Chronos baselines with sample-based tokenization, and $10.1\,\%$ for patch-based Chronos models, supporting the significance of our results.

For our main experiments, where we compare models of different sizes, we report the end-to-end **inference time** as it is of high practical interest. This also includes tokenization and detokenization overhead, which is negligible in practice with $< 0.5\,\%$ in runtime of our fastest models. We use the same Nvidia A6000 GPU for profiling with 2 warm-up and 2 measurement runs per batch to achieve inference time standard deviations $< 2\,\%$.

Regarding efficiency measures, we additionally report the **compression** at time series level of our tokenizer. This is a hardware- and model-independent measure and the metric most related to our work. Needing to process fewer tokens or requiring fewer autoregressions directly translates to improvements in inference time of models, which, however, is a hardware-dependent measure.

Finally, we suggest executing tokenization and detokenization as pipelined pre- and postprocessing operations on the CPU. This way, the minimal tokenization overhead does not affect throughput at all as model execution on the GPU is the limiting factor.

*Table 10.* MSE standard deviations for small models with our medium compression tokenizer and Chronos baselines on 5 datasets computed from 6 random seeds.

| Dataset | Chronos | | Ours |
|---|---|---|---|
| | sample-based | patch | |
| ETTh1 | 0.009 | 0.022 | 0.130 |
| ETTm1 | 0.036 | 0.134 | 0.051 |
| Weather | 0.017 | 0.026 | 0.011 |
| Electricity | 0.007 | 0.026 | 0.003 |
| Traffic | 0.228 | 0.027 | 0.017 |

**Computational effort**   Building the vocabulary of our tokenizer is an iterative process. Computationally, this is rather cheap and we execute it on a single core of an Intel Xeon w5-3435X CPU. For our medium compression tokenizer and $100\,000$ time series with a total of $100\,\mathrm{M}$ samples, vocabulary generation only takes 3.8 hours utilizing $1.2\,\mathrm{GB}$ of CPU memory. Analytically computing the conditional decoding distributions is even faster and generally takes under 10 seconds. For training the T5 models, we utilize Nvidia H100 GPUs. In total, we train 36 foundation models of different sizes and with different motif-based tokenizers. We estimate the computational effort to reproduce our experiments in table 11. Please note that we reuse previously trained tokenizers and models in most of our experiments.

*Table 11.* Computational effort to reproduce our experiments.

| Experiment | Device | Hours |
|---|---|---|
| **Tokenizer** | | |
| low | CPU | 9.1 |
| medium | CPU | 3.8 |
| high | CPU | 2.5 |
| **T5 models** | | |
| Chronos baselines | GPU | 4350 |
| Main experiments | GPU | 4800 |
| Vocabulary complexity and generalization | GPU | 2240 |
| Training dataset size | GPU | 2880 |
| MSE standard deviations | GPU | 4050 |

# C. Results

Here, we show additional experiments and results.

## C.1. Preprocessing strategies

We conduct new experiments exploring different preprocessing strategies before discretizing a time series. Each of these methods features different tradeoffs between signal preservation and noise rejection. The first derivative of a time series $z$ removes its offset, potentially yielding more similar motifs. However, derivatives generally introduce noise. To counter this, we utilize Gaussian kernels to smooth the time series. We further employ window-based normalization. Besides uniform distributions for discretization, we experiment with Gaussian distributions and the precise data distribution $P(\mathcal{D})$.
We conduct an extensive search among combinations of preprocessing strategies on $500$ tokenizers in figure 8. Uniform discretization with a different number of bins $M$ is Pareto optimal in balancing the average tokenization error $\delta_{\mathrm{avg}}$ and compression $\bar{c}$. We utilize this method throughout our paper.

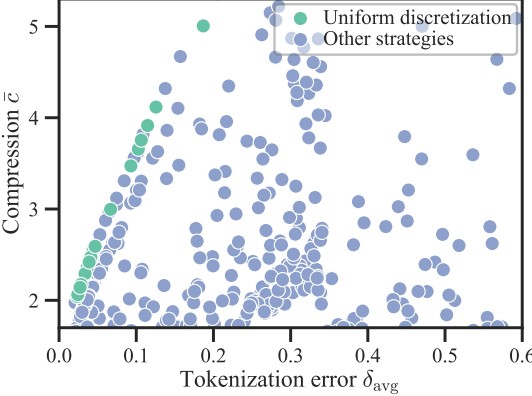

*Figure 8.* Comparison of uniform discretization with other tokenization preprocessing strategies, including derivatives, Gaussian smoothing, window-based normalization, and Gaussian- and data-distribution-based discretization.

## C.2. Efficiency improvements of adaptive tokenization

In table 12 and figure 9, we report full, non-aggregated results comparing our motif-based tokenization with Chronos foundation models that tokenize every sample. We conduct additional experiments to compare our motif-based tokenization with non-foundation models tokenizing every sample. Finally, we isolate the effects of discretization and temporal motif representation and investigate different input lengths.

*Table 12.* Comparison of MSE and inference time of Chronos models and our low, medium, and high compression tokenizers with conditional decoding (cd) and without on 7 datasets and 5 model sizes. **Best** MSE in bold.

| Dataset | Model size | Chronos | | Ours$_{low}$ | | | Ours$_{medium}$ | | | Ours$_{high}$ | | |
|---|---|---|---|---|---|---|---|---|---|---|---|---|
| | | MSE | time | MSE | MSE$^{cd}$ | time | MSE | MSE$^{cd}$ | time | MSE | MSE$^{cd}$ | time |
| ETTh1 | tiny | 0.744 | 0.031 s | 0.854 | 0.617 | 0.022 s | 0.720 | 0.545 | 0.016 s | 0.881 | **0.540** | 0.016 s |
| | mini | 0.736 | 0.061 s | 0.803 | 0.599 | 0.044 s | 0.585 | **0.507** | 0.035 s | 0.758 | 0.520 | 0.031 s |
| | small | 0.741 | 0.094 s | 0.656 | 0.565 | 0.067 s | 0.669 | **0.500** | 0.057 s | 0.686 | 0.512 | 0.051 s |
| | base | 0.759 | 0.305 s | 0.602 | 0.525 | 0.204 s | 0.554 | 0.465 | 0.175 s | 0.528 | **0.463** | 0.165 s |
| | large | 0.717 | 0.867 s | 0.530 | 0.487 | 0.575 s | 0.527 | 0.461 | 0.507 s | 0.517 | **0.459** | 0.456 s |
| ETTm1 | tiny | 1.138 | 0.031 s | 1.044 | 0.637 | 0.020 s | 1.063 | 0.619 | 0.016 s | 0.904 | **0.585** | 0.014 s |
| | mini | 1.105 | 0.061 s | 1.031 | 0.644 | 0.041 s | 1.018 | **0.560** | 0.033 s | 1.017 | 0.589 | 0.030 s |
| | small | 1.004 | 0.094 s | 0.934 | 0.609 | 0.064 s | 0.826 | **0.495** | 0.054 s | 0.933 | 0.520 | 0.050 s |
| | base | 1.061 | 0.305 s | 0.887 | 0.590 | 0.184 s | 0.759 | 0.473 | 0.165 s | 0.660 | **0.460** | 0.152 s |
| | large | 1.084 | 0.867 s | 0.764 | 0.569 | 0.540 s | 0.784 | 0.487 | 0.488 s | 0.637 | **0.449** | 0.438 s |
| Weather | tiny | 0.313 | 0.031 s | 0.525 | 0.331 | 0.015 s | 0.406 | 0.290 | 0.012 s | 0.338 | **0.284** | 0.013 s |
| | mini | 0.297 | 0.061 s | 0.482 | 0.305 | 0.032 s | 0.324 | **0.257** | 0.026 s | 0.313 | 0.280 | 0.027 s |
| | small | 0.265 | 0.094 s | 0.463 | 0.298 | 0.050 s | 0.344 | 0.250 | 0.046 s | 0.290 | **0.238** | 0.044 s |
| | base | 0.266 | 0.305 s | 0.535 | 0.316 | 0.138 s | 0.307 | **0.241** | 0.140 s | 0.273 | 0.258 | 0.132 s |
| | large | 0.269 | 0.867 s | 0.492 | 0.316 | 0.418 s | 0.293 | 0.242 | 0.405 s | 0.251 | **0.236** | 0.367 s |
| Electricity | tiny | 0.375 | 0.031 s | 0.246 | 0.228 | 0.021 s | 0.241 | **0.223** | 0.016 s | 0.245 | 0.224 | 0.015 s |
| | mini | 0.301 | 0.061 s | 0.200 | 0.192 | 0.043 s | 0.198 | **0.186** | 0.033 s | 0.199 | 0.187 | 0.027 s |
| | small | 0.261 | 0.094 s | 0.176 | 0.169 | 0.066 s | 0.170 | **0.161** | 0.056 s | 0.185 | 0.173 | 0.048 s |
| | base | 0.222 | 0.305 s | 0.167 | 0.159 | 0.203 s | 0.165 | **0.157** | 0.163 s | 0.166 | 0.158 | 0.148 s |
| | large | 0.233 | 0.867 s | 0.154 | 0.148 | 0.569 s | 0.150 | **0.144** | 0.471 s | 0.158 | 0.151 | 0.397 s |
| Traffic | tiny | 4.682 | 0.031 s | 0.805 | 0.756 | 0.021 s | 0.825 | 0.762 | 0.017 s | 0.755 | **0.721** | 0.015 s |
| | mini | 3.751 | 0.061 s | 0.716 | 0.684 | 0.042 s | 0.682 | **0.648** | 0.033 s | 0.680 | 0.650 | 0.027 s |
| | small | 2.722 | 0.094 s | 0.693 | 0.646 | 0.065 s | 0.659 | **0.617** | 0.056 s | 0.646 | 0.627 | 0.047 s |
| | base | 3.413 | 0.305 s | 0.686 | 0.631 | 0.201 s | 0.630 | **0.585** | 0.163 s | 0.631 | 0.608 | 0.143 s |
| | large | 2.717 | 0.867 s | 0.688 | 0.628 | 0.576 s | 0.613 | 0.576 | 0.474 s | 0.591 | **0.574** | 0.386 s |
| Solar | tiny | 1.387 | 0.031 s | 1.633 | 0.729 | 0.014 s | 1.348 | **0.595** | 0.013 s | 1.117 | 0.613 | 0.013 s |
| | mini | 1.270 | 0.061 s | 1.442 | 0.715 | 0.029 s | 0.963 | 0.521 | 0.028 s | 0.767 | **0.491** | 0.027 s |
| | small | 1.358 | 0.094 s | 1.316 | 0.720 | 0.045 s | 0.845 | **0.449** | 0.048 s | 0.677 | 0.450 | 0.045 s |
| | base | 1.311 | 0.305 s | 1.265 | 0.710 | 0.123 s | 0.870 | **0.458** | 0.131 s | 0.688 | 0.491 | 0.125 s |
| | large | 1.319 | 0.867 s | 1.210 | 0.715 | 0.388 s | 0.751 | 0.419 | 0.419 s | 0.493 | **0.371** | 0.369 s |
| Fev-bench | tiny | 1.562 | 0.031 s | 1.223 | 0.892 | 0.021 s | 1.195 | 0.862 | 0.017 s | 1.093 | **0.815** | 0.015 s |
| | mini | 1.499 | 0.061 s | 1.155 | 0.864 | 0.043 s | 1.137 | 0.835 | 0.034 s | 1.089 | **0.799** | 0.029 s |
| | small | 1.584 | 0.094 s | 1.119 | 0.839 | 0.066 s | 1.082 | 0.807 | 0.094 s | 1.069 | **0.778** | 0.049 s |
| | base | 1.577 | 0.305 s | 1.095 | 0.832 | 0.189 s | 1.071 | 0.780 | 0.164 s | 1.038 | **0.769** | 0.141 s |
| | large | 1.489 | 0.867 s | 1.060 | 0.802 | 0.556 s | 1.033 | 0.778 | 0.475 s | 1.006 | **0.756** | 0.391 s |

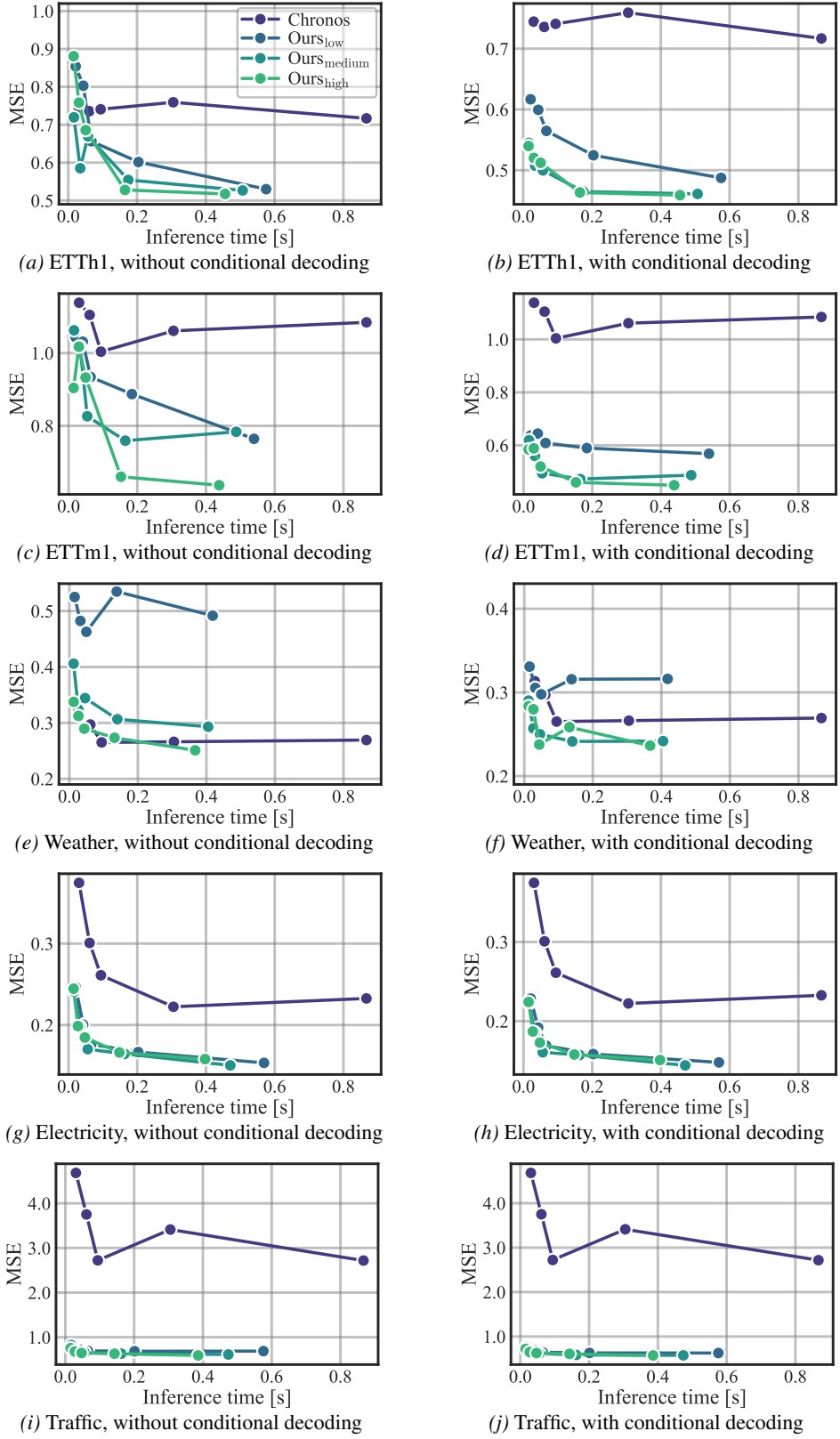

*Figure 9.* Comparison of our motif-based tokenization with and without conditional decoding with Chronos models tokenizing every sample during zero-shot evaluation on 7 datasets and 5 model sizes.

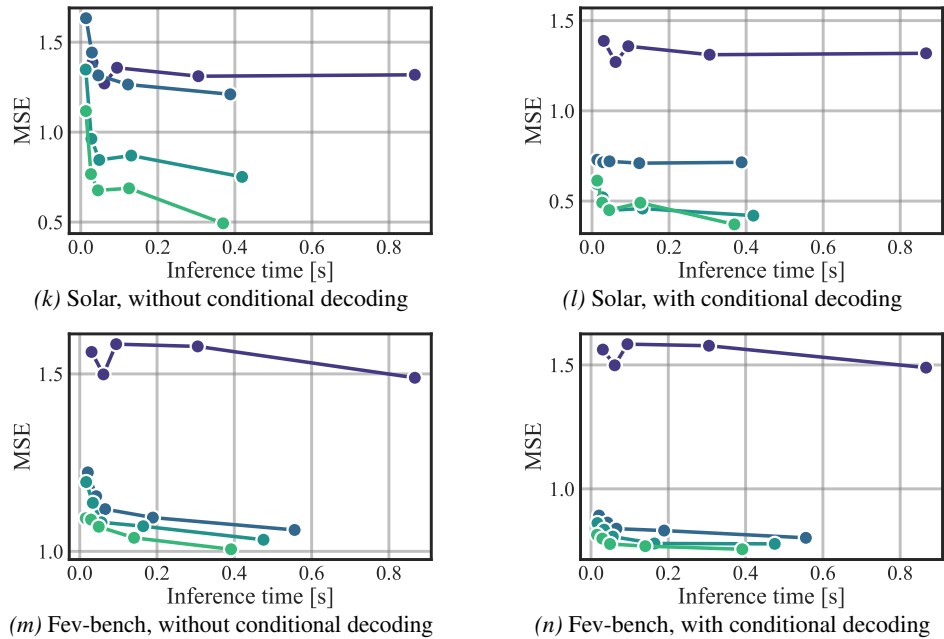

*(k)* Solar, without conditional decoding

*(l)* Solar, with conditional decoding

*(m)* Fev-bench, without conditional decoding

*(n)* Fev-bench, with conditional decoding

*Figure 9.* (continued)

**Non-foundation models** Here, we compare our motif-based tokenization in T5 foundation models with non-foundation models that are specifically trained on the ETTh1, ETTm1, Weather, Electricity, and Traffic datasets. We utilize common time series architectures including Autoformer (Wu et al., 2021), FEDformer (Zhou et al., 2022), Informer (Zhou et al., 2021), Non-stationary (Liu et al., 2022b), and vanilla transformers (Vaswani et al., 2017). These models extract tokens as a multivariate slice for every time step. For comparison with our motif-based tokenization, we utilize the results of Götz et al. (2025) for 2 layer models in table 13. The authors forecast 96 time series samples from 192 context tokens.

In 19 out of 25 cases, our foundation model in a zero-shot setting outperforms the specifically trained models in forecasting quality.

*Table 13.* Comparison of our motif-based tokenization with conditional decoding (cd) and without in zero-shot foundation models with non-foundation models that tokenize every sample, based on forecasting quality (MSE). We highlight values that are  worse  than our method.

| Dataset | Ours | Ours[cd] | Autoformer | FEDformer | Informer | Non-stationary | Transformer |
|---|---|---|---|---|---|---|---|
| ETTh1 | 0.52 | 0.46 | 0.42 | 0.38 | 0.87 | 0.55 | 0.75 |
| ETTm1 | 0.64 | 0.45 | 0.44 | 0.36 | 0.65 | 0.42 | 0.52 |
| Weather | 0.25 | 0.24 | 0.28 | 0.27 | 0.35 | 0.19 | 0.25 |
| Electricity | 0.15 | 0.14 | 0.18 | 0.20 | 0.30 | 0.17 | 0.25 |
| Traffic | 0.59 | 0.57 | 0.63 | 0.59 | 0.68 | 0.60 | 0.66 |

**Effects of discretization and temporal motifs** Our motif-based tokenization consists of two steps: Quantizing the time series into a sequence of discrete symbols and compressing this sequence using a vocabulary of temporal motifs. To isolate the contributions of discretization and motif representation, we train models from tiny to large with the same number of quantization bins ($M = 37$) as our medium compression tokenizer, but without applying motif discovery.

Our results in table 14 show that models relying solely on discretization perform worse than those incorporating motif-based representations, except for the Weather dataset. However, they still outperform Chronos baselines. The improved MSE of the motif-based approach can be attributed to the fact that longer motifs provide the model with higher-level building blocks, making sequence prediction easier and more accurate (this is supported by our results that longer motifs are associated with smaller MSE, see section 5.5). Moreover, our motif representation also compresses temporal patterns, resulting in efficiency gains, which is a main motivation for our work. Please note that the Weather dataset features extraordinarily high average compressions of 23.15 in table 4, which may be the reason for the decreased accuracy. However, here the motif-based model is substantially more efficient.

*Table 14.* Forecasting quality (MSE) on 5 evaluation datasets for models from tiny to large, with $M = 37$ quantization bins, trained with and without motif representations. **Best** in bold.

| Dataset | Discretization | | | | | Motifs |
|---|---|---|---|---|---|---|
| | tiny | mini | small | base | large | |
| ETTh1 | 0.611 | 0.608 | 0.600 | 0.625 | 0.626 | **0.526** |
| ETTm1 | 1.035 | 0.911 | 0.928 | 0.887 | 0.918 | **0.759** |
| Weather | 0.231 | 0.229 | 0.220 | 0.222 | **0.215** | 0.293 |
| Electricity | 0.252 | 0.223 | 0.211 | 0.196 | 0.189 | **0.150** |
| Traffic | 1.099 | 1.135 | 1.195 | 1.122 | 1.184 | **0.613** |

**Effects of input length** In our main experiments, we compare sample-based tokenization in Chronos models with our motif-based tokenizer. We use 128 context tokens in both cases to ensure an identical compute budget for the transformer encoder, such that any efficiency gains arise solely from compressed token generation in the decoder. As a consequence, the two tokenization schemes induce different effective input lengths. To disentangle the effect of input length from token expressivity, we additionally compare sample-based tokenization to our medium compression tokenizer without conditional decoding using the same input length of 384 time series samples. Note that all comparisons to patch-based tokenizers also use this context length.

Table 15 confirms that the observed improvements in predictive performance result from the increased expressiveness of motif-based tokenization rather than from longer inputs. Due to the larger input length, these Chronos models are on average $1.24\times$ slower than those reported throughout. As noted earlier, the Weather dataset exhibits exceptionally high average compression ratios of 23.15 in table 4, which may explain the reduced accuracy. However, the motif-based model remains substantially more efficient in this setting.

*Table 15.* Forecasting quality (MSE) on 5 evaluation datasets for Chronos models and our motif-based tokenization using the same input length of 384 time series samples. **Best** in bold.

| Dataset | Chronos | | | | | Motifs |
|---|---|---|---|---|---|---|
| | tiny | mini | small | base | large | |
| ETTh1 | 0.697 | 0.713 | 0.693 | 0.630 | 0.636 | **0.527** |
| ETTm1 | 1.059 | 0.999 | 0.862 | 0.787 | 0.764 | **0.759** |
| Weather | 0.263 | **0.223** | 0.241 | 0.233 | 0.253 | 0.293 |
| Electricity | 0.312 | 0.266 | 0.208 | 0.184 | 0.172 | **0.150** |
| Traffic | 3.296 | 2.793 | 2.549 | 1.934 | 1.937 | **0.613** |

## C.3. Conditional decoding

In this section, we provide full results on data- and model-dependent conditional decoding trained on the models' predictions in figure 10. We further explore conditional decoding in a data- and model-independent setting. Additionally, we investigate higher-order conditional decoding schemes and demonstrate that conditional decoding can also enhance the predictive performance of patch-based models.

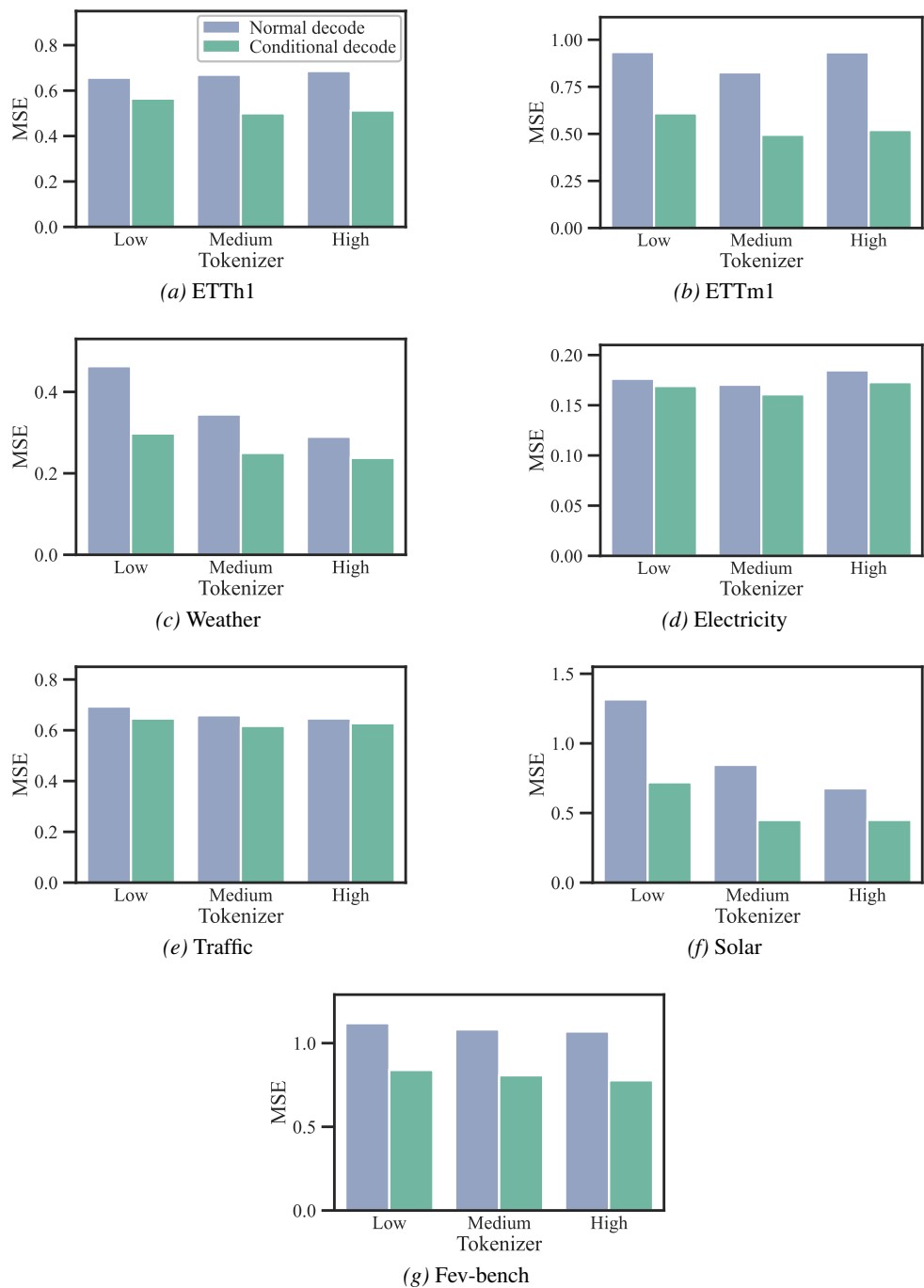

*Figure 10.* Conditional decoding improves forecasting quality for 3 tokenizers in small models on 7 datasets.

**Data- and model-independent conditional decoding**  Here, we investigate conditional decoding in a data- and model-independent setting to universally improve the forecasting quality of foundation models. To this end, we train conditional decoding to dequantize quantized time series $z' = \mathbf{q}_\Omega(z)$. Here, conditional decoding is model-independent and can only mitigate the quantization error as this is the only error introduced. This is why we report MSE improvements relative to the average quantization error $\delta_{\text{avg}}$ on the respective evaluation datasets. We utilize the Chronos dataset for training and 5 datasets for zero-shot evaluation. We demonstrate conditional decoding on our 3 tokenizers and small models.

In 14 out of 15 settings in table 16, conditional decoding improves forecasting quality. On ETTm1, it mitigates up to 96.9 % of the quantization error of our tokenizer with high compression. This enables us to build tokenizers with even higher compression and quantization error, as it can be effectively recovered.

*Table 16.* Conditional decoding in data- and model-independent setting recovers the quantization errors of our 3 tokenizers by varying degrees on 5 datasets.

| Dataset | Compression | | |
|---|---|---|---|
| | low | medium | high |
| ETTh1 | 22.0 % | 21.3 % | 49.4 % |
| ETTm1 | 61.1 % | 16.1 % | 96.9 % |
| Weather | 87.6 % | 38.9 % | 25.0 % |
| Electricity | 13.5 % | 23.0 % | 21.4 % |
| Traffic | 0.0 % | 0.7 % | 1.7 % |

**Higher-order conditional decoding**  We propose conditional decoding as a post-hoc optimization method for transforming discrete tokens back into continuous space. Throughout our experiments, we explore very lightweight conditional decoding using the first-order Markov assumption. Here, we investigate longer look-back windows, i.e., conditioning on the 2 or 3 previous samples. These higher-order models with exponentially more parameters may lead to better performance in special cases but might also hinder generalization or overfit. We utilize small models, our medium compression tokenizer, and data- and model-dependent conditional decoding (see section 5.3) for this experiment.

Our results in table 17 demonstrate that higher-order conditional decoding only marginally improves predictive performance. While first-order conditional decoding has $M^2 = 1369$ parameters, parameter count is substantially increased for third-order models $M^4 = 1\,874\,161$. This demonstrates the effectiveness of our first-order method.

*Table 17.* Comparison of conditional decoding with different Markov orders on 7 datasets. **Best** in bold.

| Dataset | MSE | MSE$^{\text{cd}}$ | | |
|---|---|---|---|---|
| | | first-order | second-order | third-order |
| ETTh1 | 0.669 | **0.500** | 0.503 | 0.516 |
| ETTm1 | 0.826 | 0.495 | 0.489 | **0.486** |
| Weather | 0.344 | 0.250 | 0.246 | **0.242** |
| Electricity | 0.170 | **0.161** | **0.161** | 0.163 |
| Traffic | 0.659 | **0.617** | **0.617** | 0.623 |
| Solar | 0.845 | 0.449 | 0.447 | **0.445** |
| Fev-bench | 1.082 | 0.807 | 0.775 | **0.738** |
| Average | 0.656 | 0.468 | 0.463 | 0.459 |

**Conditional decoding for patch-based models**   We demonstrate a direct application of conditional decoding to post-hoc refine the predictions of patch-based models from the literature. To this end, we first quantize the predictions of patch-based models to reduce noise. In a second step, we transform them back into time series representations using data- and model-dependent conditional decoding (see section 5.3). Here, conditional decoding may simultaneously act as a lightweight domain adaptation, promoting dataset-specific patterns.

Our results in table 18 show that conditional decoding improves the forecasting quality of patch-based models by $5\%$ on average, without modifying or retraining the model itself. This post-hoc application makes our tokenization directly compatible with current continuous-embedding architectures in the literature.

*Table 18.* Tokenization of predictions of patch-based literature models with subsequent conditional decoding post-hoc improves prediction quality by $5.0\%$ on average without modifying or retraining the model itself.

| Dataset | MOMENT | | | Moirai | | | Time-MoE | | LightGTS |
|---|---|---|---|---|---|---|---|---|---|
| | small | base | large | small | base | large | base | large | |
| ETTh1 | 0.0% | 0.0% | 0.0% | 0.0% | 0.0% | 0.0% | 0.2% | 0.4% | 0.8% |
| ETTm1 | 0.7% | 0.0% | 0.0% | 5.8% | 3.2% | 1.3% | 0.0% | 0.8% | 10.9% |
| Weather | 0.4% | 0.0% | 0.0% | 1.6% | 0.0% | 25.7% | 1.4% | 4.3% | 0.0% |
| Electricity | 2.6% | 0.0% | 0.0% | 2.3% | 0.0% | 0.0% | 5.6% | 4.9% | 6.2% |
| Traffic | 3.5% | 2.4% | 0.4% | 0.0% | 0.0% | 0.0% | 6.0% | 4.3% | 0.0% |
| Solar | 2.2% | 1.6% | 0.0% | 30.8% | 26.8% | 33.5% | 26.5% | 17.7% | 6.2% |
| Fev-bench | 9.4% | 8.4% | 6.6% | 6.7% | 6.8% | 5.8% | 14.2% | 13.3% | 0.0% |
| Average | 2.7% | 1.8% | 1.0% | 6.7% | 5.3% | 9.5% | 7.7% | 6.5% | 3.4% |

## C.4. Adaptive compression of diverse time series

Here, we present the complete results of our investigations on adaptive compression of our motif-based tokenization approach. In figure 11, we explore variable compression within the same dataset and show tokenized time series for visual inspection in figure 13. We further investigate relations between input and generation compression. Finally, we list compression rates of patch-based literature models as a reference.

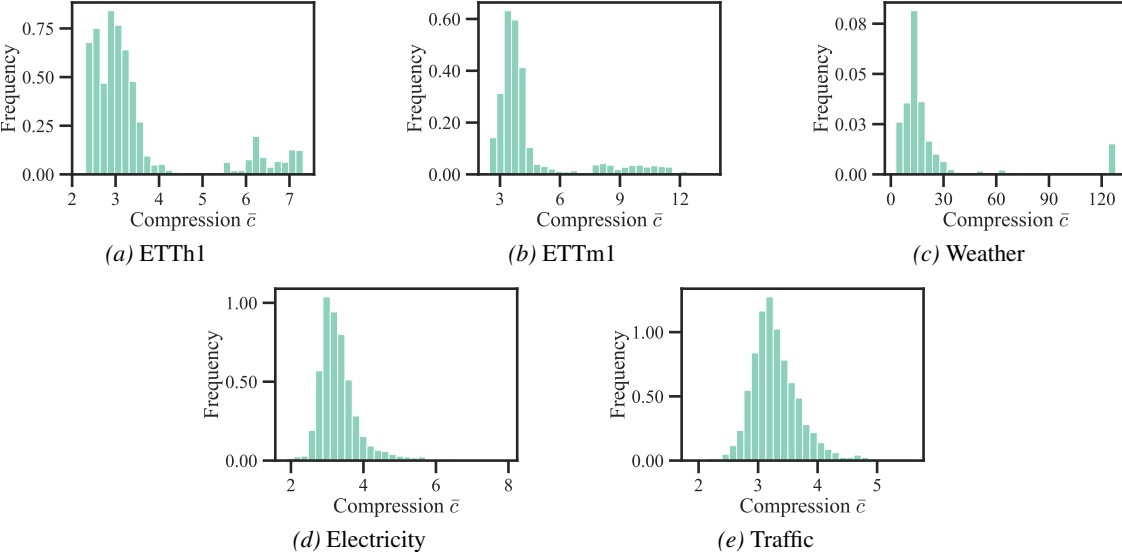

*Figure 11.* Histograms showing variable compressions of our medium tokenizer within 5 datasets.

**Input and generation compression** We conduct additional experiments to explore relations between input compression and the model's generations. The model can either predict long motifs with high compression directly or sequences of their shorter components during autoregressive generation, as we describe in section 5.7. Therefore, we expect a greater input compression $\bar{c}_{\text{in}}$ compared to generation compression $\bar{c}_{\text{out}}$. We utilize different tokenizers in small models for this experiment. In line with our hypothesis, we find correlations between input and generation compression on all 5 datasets in figure 12. More complex input tokens generally benefit the prediction of longer motifs.

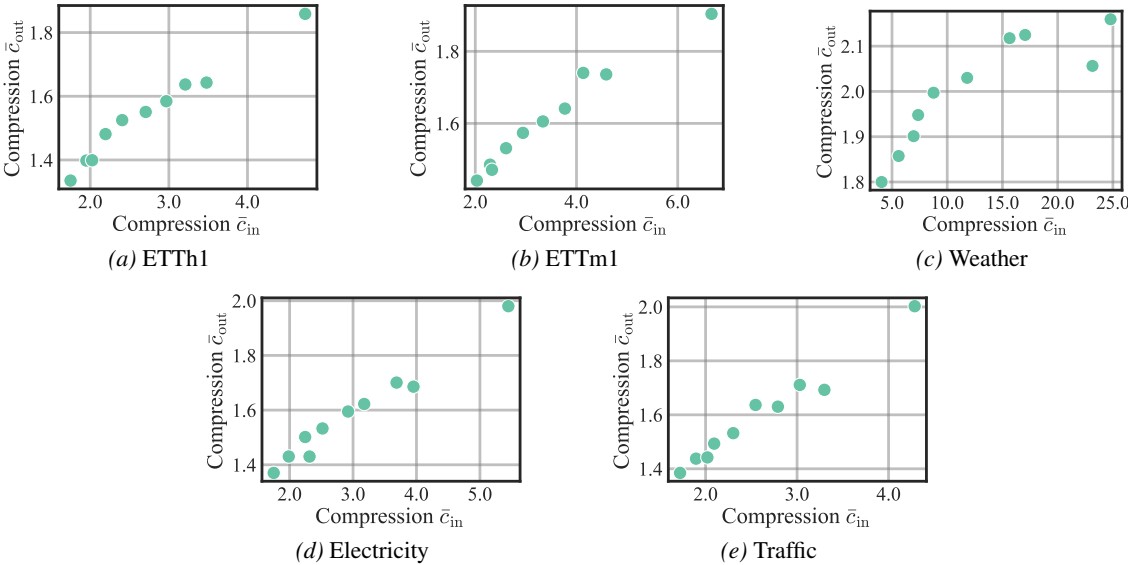

*Figure 12.* Comparison of input and generation compression of small models and multiple tokenizers on 5 datasets. Please note that efficiency gains in tables 2 and 12 and figures 3 and 9 relate to the more conservative generation compression.

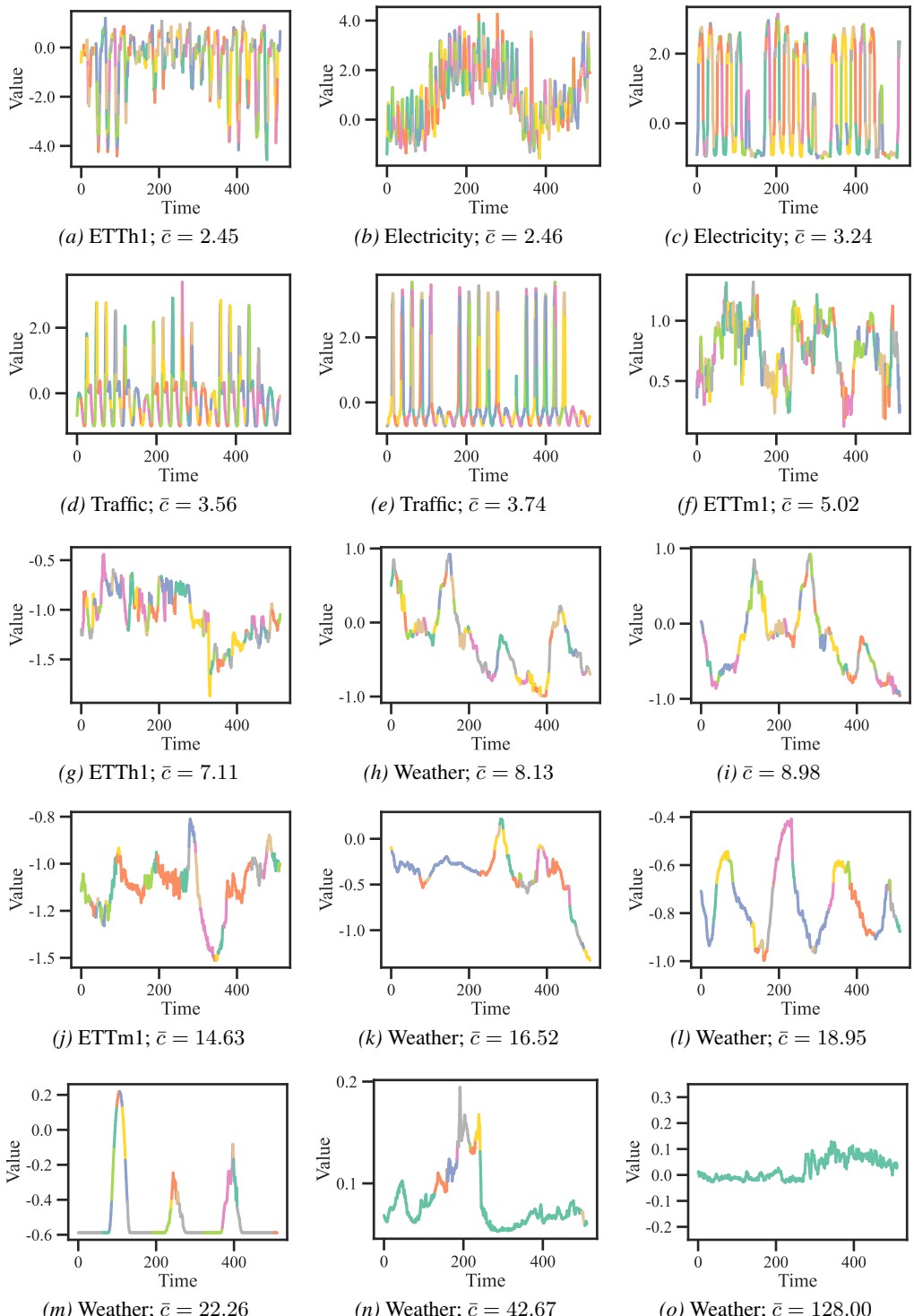

*Figure 13.* Our medium tokenizer exploits periodically recurring motifs and compresses time series adaptively depending on pattern complexity on 5 datasets. Specifically, **(o)** highlights the noise rejection ability of discretization.

**Compression rates of patch-based models** In table 19 we list compression rates of patch-based models in recent literature, resulting from different patch length and stride combinations. For some works that experiment with multiple patch lengths, we show the authors' preferred values.

*Table 19.* Compression rates of patch-based literature models.

| Architecture | Compression $\bar{c}$ |
|---|---:|
| SDformer (Chen et al., 2024) | 2, 4 |
| TOTEM (Talukder et al., 2024) | 4 |
| MOMENT (Goswami et al., 2024) | 8 |
| PatchTST (Nie et al., 2023) | 8 |
| TimeXer (Wang et al., 2024) | 16 |
| UniTS (Gao et al., 2024) | 16 |
| Sundial (Liu et al., 2025) | 16 |
| Moirai-MoE (Liu et al., 2024a) | 16 |

## C.5. Vocabulary complexity and generalization

In figure 14 and table 20, we provide full results of our investigations on token occurrence. We offer additional insights into the hierarchy of motifs and the vocabulary generation process.

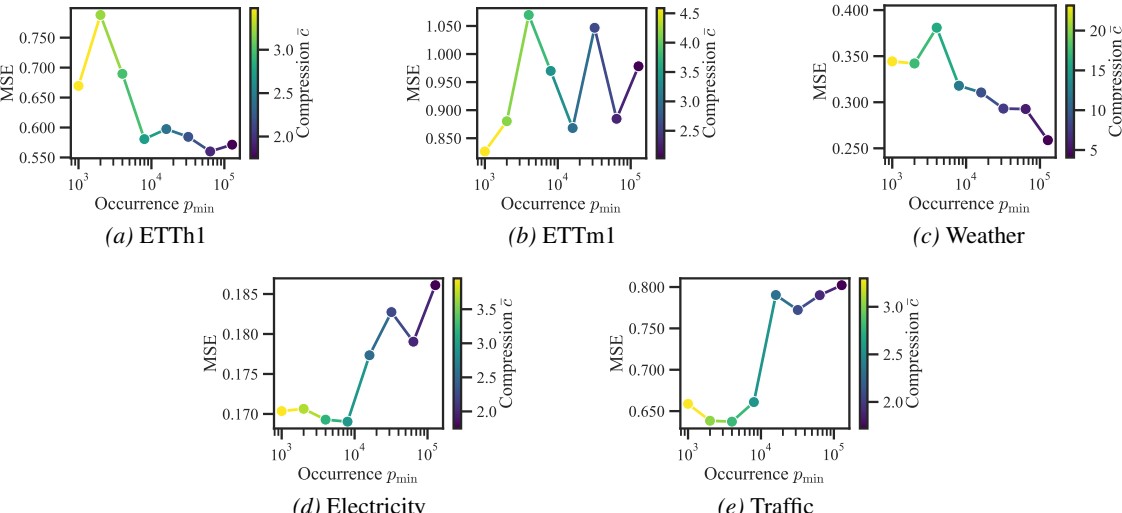

*(a)* ETTh1         *(b)* ETTm1         *(c)* Weather

*(d)* Electricity         *(e)* Traffic

*Figure 14.* Varying token occurrence $p_{\min}$ influences forecasting quality for small models on 5 datasets.

*Table 20.* Tokenizers on the Chronos dataset with different token occurrence, vocabulary size, and compression.

| $p_{\min}$ | $|\mathcal{V}|$ | $\bar{c}$ |
|---|---|---|
| 1000 | 1675 | 3.18 |
| 2000 | 993 | 2.95 |
| 4000 | 604 | 2.73 |
| 8000 | 373 | 2.50 |
| 16 000 | 237 | 2.29 |
| 32 000 | 158 | 2.08 |
| 64 000 | 108 | 1.86 |
| 128 000 | 78 | 1.66 |

**Hierarchy of motifs**   Motif-based tokenization utilizes a vocabulary of hierarchical patterns. Here, we explore the hierarchy of motifs evolving with more complex vocabularies. To this end, we vary the minimum occurrence threshold $p_{\min}$. Naturally, a lower occurrence threshold results in larger vocabularies. These vocabularies exhibit more complex patterns generated by a greater number of recursive merges in figure 15. Due to its hierarchical structure, motif length grows exponentially with vocabulary depth, enabling substantial compression.

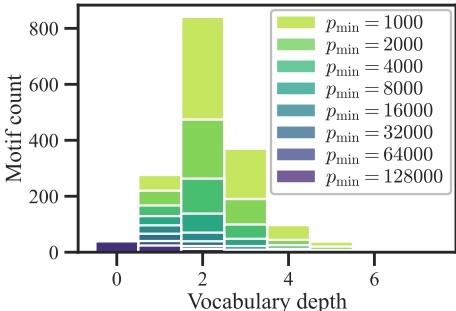

*Figure 15.* Motif hierarchy for vocabularies of different complexity.

**Vocabulary generation process**   Here, we further highlight the influence of quantization granularity and token occurrence on compression and vocabulary complexity. In figure 16, we show the iterative process of finding longer motifs with higher compression $\bar{c}$ during vocabulary generation of $\Psi$. These more complex motifs, however, are more specialized and occur less often ($p_{\min}$). A lower number of quantization bins $M$ results in smaller, less data-specific vocabularies with higher compression.

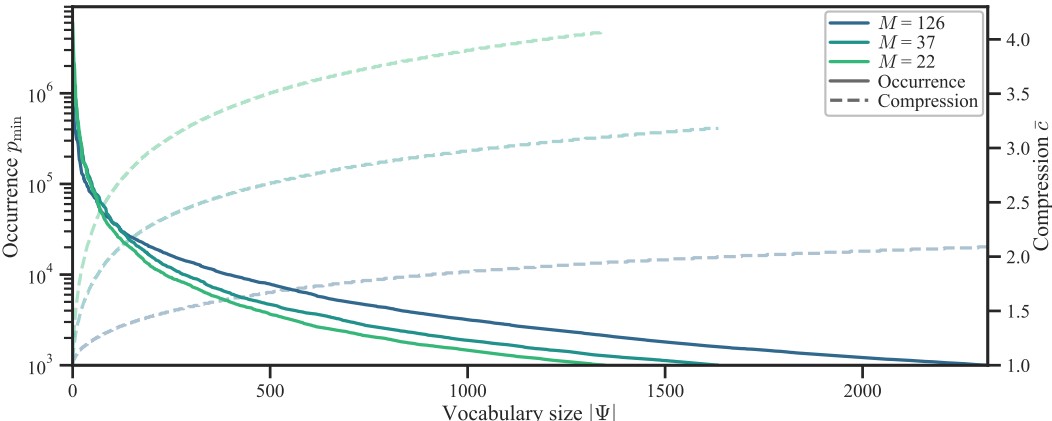

*Figure 16.* Influence of quantization bins $M$ and token occurrence $p_{\min}$ on vocabulary size $|\Psi|$ and compression $\bar{c}$ for tokenizers on the Chronos dataset.

### C.6. Robustness to noise, non-stationary time series, and transients

Robustness to noise is of high relevance when processing real-world time series. Further, the changing distribution of non-stationary time series or extreme values might hinder effective motif-based tokenization. Here, we explore our tokenizer's generalization to noise, distribution shifts, and transients in more detail.

**Robustness to noise**  To explore the noise rejection capability of motif-based tokenization, we injected different levels of Gaussian noise into the raw input sequence before tokenization. We compare our high compression tokenizer in a small model to the respective Chronos baseline.

Our results in figure 17 show that our motif-based tokenization substantially outperforms Chronos models on noisy data. Further, its noise resistance is more predictable. We argue that our method is more robust to noise due to its coarser quantization granularity. At the same time, our more expressive motif representation mitigates the larger discretization error. Note that adding noise with up to $\sigma = 0.3$ to our normalized input data with unit standard deviation is a severe disturbance.

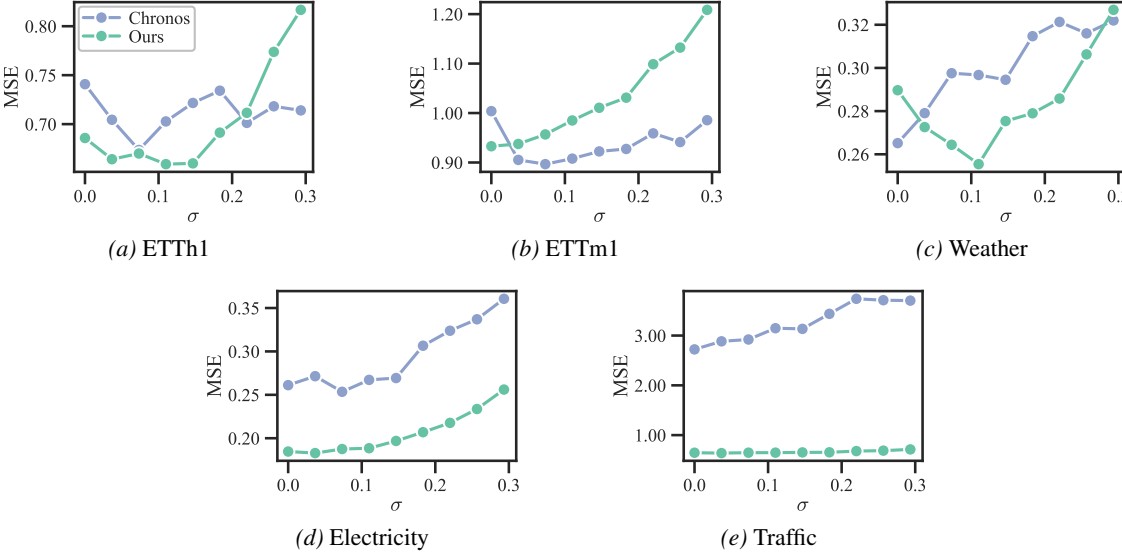

*Figure 17.* Resistance of Chronos models and our high compression tokenizer to Gaussian noise with standard deviation $\sigma$ on 5 datasets.

**Generalization to non-stationary data**  In practice, trends on long non-stationary time series might hinder effective motif encoding. To explore this, we introduce linear and exponential trends into our evaluation datasets. We utilize our high compression tokenizer in a small model and the corresponding Chronos baseline for this experiment.

In figure 18, our method shows a similar robustness to non-stationary data compared to the Chronos baseline, even for large trends. We conclude that our motif-based tokenization is well applicable to non-stationary time series and long sequences. Note that applying trends with up to $|\alpha| = 0.5$ to our normalized data is a large disturbance.

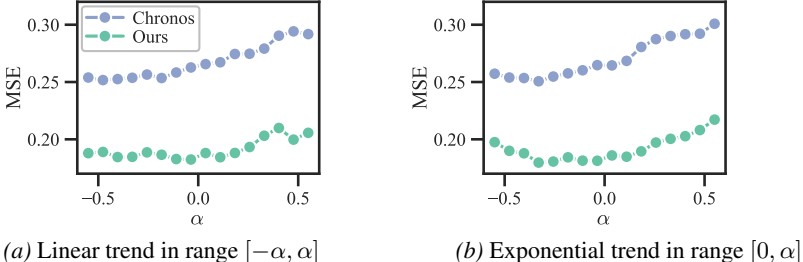

*Figure 18.* Generalization of Chronos models and our high compression tokenizer to non-stationary time series with linear and exponential trends on the Electricity dataset.

**Robustness to transients** Extreme values might occur in real-world time series. Here, we explore the robustness of our motif-based tokenization to outliers. To this end, we augment the input time series such that every sample has a $1\%$ probability to be a positive or negative transient with amplitude 3. For our normalized time series with unit standard deviation, this is a severe disturbance. We analyze our tokenizer with medium compression and models in size small on the Electricity dataset.

While Chronos models cannot effectively handle extreme values, our motif-based tokenization is substantially more robust to outliers as our results in table 21 show. The MSE of our models increases by only $19.4\%$, compared to $52.5\%$ for Chronos models. When encountering unknown patterns that are not in our tokenizer's motif vocabulary, such as transients, our tokenizer falls back to sample-based tokenization, as illustrated in figure 19. This fallback ensures that motif-based tokenization cannot overlook individual samples by design. Consequently, compression is slightly reduced by $14.4\%$ when outliers are introduced.

Note that within the tokenization range, the same maximum quantization error applies regardless of whether a sample is common or an outlier, as described in section 3.1.

*Table 21.* MSE and compression $\bar{c}$ for Chronos and our medium compression tokenizer on the Electricity dataset with and without transient augmentation.

| Augmentation | Chronos | Ours | |
| --- | --- | --- | --- |
| | MSE | MSE | $\bar{c}$ |
| Without transients | 0.261 | 0.170 | 3.95 |
| With transients | 0.398 | 0.203 | 3.38 |

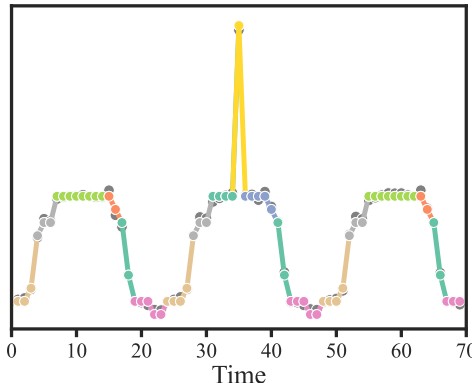

*Figure 19.* Introducing a transient (yellow) in the center period leads to local changes in tokenization of time series samples (gray) compared to the other periods on the Electricity dataset.

## C.7. Training dataset size

We vary the dataset size for building our vocabulary of motifs $\Psi$ and provide full results here. To this end, we utilize our three tokenizers with low, medium, and high compression and report vocabulary statistics in table 22 and forecasting quality in table 23. For conditional decoding, we observe similar behavior when estimating conditional distributions $\hat{\Omega}$ in table 24. Here, a larger subset also leads to best representations.

*Table 22.* Vocabulary statistics for 3 tokenizers trained on Chronos subsets of varying sizes $N$.

| Compression | $N = 1\,\text{k}$ | | $N = 10\,\text{k}$ | | $N = 100\,\text{k}$ | | $N = 1\,\text{M}$ | |
|---|---|---|---|---|---|---|---|---|
| | $|\mathcal{V}|$ | $\bar{c}$ | $|\mathcal{V}|$ | $\bar{c}$ | $|\mathcal{V}|$ | $\bar{c}$ | $|\mathcal{V}|$ | $\bar{c}$ |
| low | 2789 | 2.11 | 2461 | 2.08 | 2445 | 2.08 | 2441 | 2.09 |
| medium | 1974 | 3.24 | 1707 | 3.16 | 1675 | 3.18 | 1681 | 3.18 |
| high | 1618 | 4.14 | 1392 | 4.05 | 1373 | 4.06 | 1360 | 4.06 |

*Table 23.* Forecasting quality (MSE) on 5 evaluation datasets for 3 tokenizers trained on Chronos subsets of varying sizes $N$.

| Dataset | Compression | $N = 1\,\text{k}$ | $N = 10\,\text{k}$ | $N = 100\,\text{k}$ | $N = 1\,\text{M}$ |
|---|---|---|---|---|---|
| | low | 0.712 | 0.712 | 0.656 | 0.659 |
| ETTh1 | medium | 0.562 | 0.698 | 0.669 | 0.615 |
| | high | 0.751 | 0.712 | 0.686 | 0.593 |
| | low | 0.944 | 0.919 | 0.934 | 0.913 |
| ETTm1 | medium | 0.877 | 0.898 | 0.826 | 0.857 |
| | high | 0.985 | 0.819 | 0.933 | 0.821 |
| | low | 0.473 | 0.538 | 0.463 | 0.474 |
| Weather | medium | 0.342 | 0.310 | 0.344 | 0.333 |
| | high | 0.355 | 0.333 | 0.290 | 0.307 |
| | low | 0.178 | 0.183 | 0.176 | 0.175 |
| Electricity | medium | 0.173 | 0.167 | 0.170 | 0.164 |
| | high | 0.191 | 0.176 | 0.185 | 0.178 |
| | low | 0.724 | 0.680 | 0.693 | 0.671 |
| Traffic | medium | 0.643 | 0.639 | 0.659 | 0.622 |
| | high | 0.625 | 0.621 | 0.646 | 0.620 |

*Table 24.* Influence of dataset size $N$ on estimating conditional decoding distributions $\hat{\Omega}$ for small models on the Electricity dataset.

| $N$ | MSE |
|---|---|
| Normal decoding | 0.170 |
| $1\,\text{k}$ | 0.168 |
| $10\,\text{k}$ | 0.162 |
| $100\,\text{k}$ | 0.161 |

## C.8. Learned token representations

Here, we present our comprehensive explainability analysis of the learned motif representations.

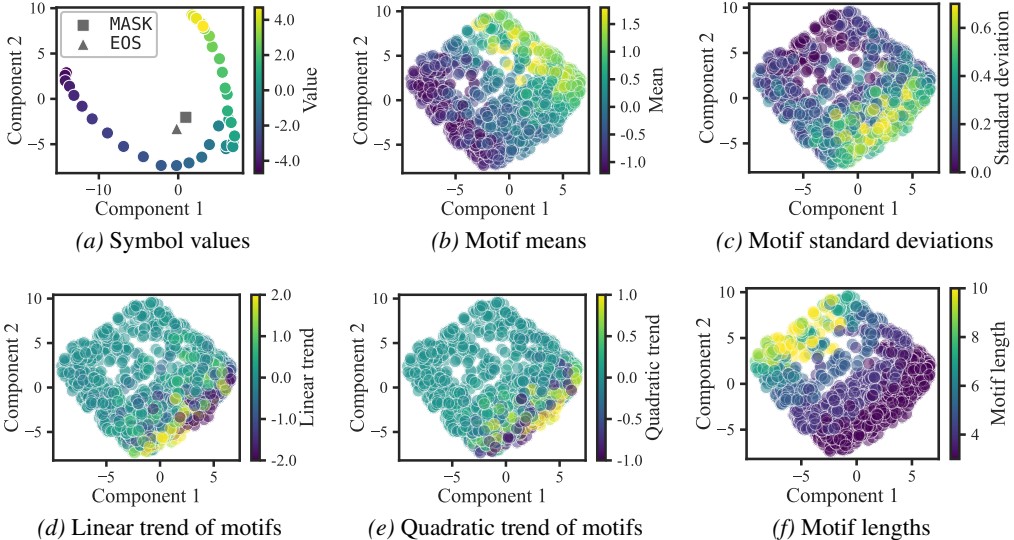

*(a)* Symbol values   *(b)* Motif means   *(c)* Motif standard deviations

*(d)* Linear trend of motifs   *(e)* Quadratic trend of motifs   *(f)* Motif lengths

*Figure 20.* Principal component analysis of token embeddings of our medium tokenizer in a small model.

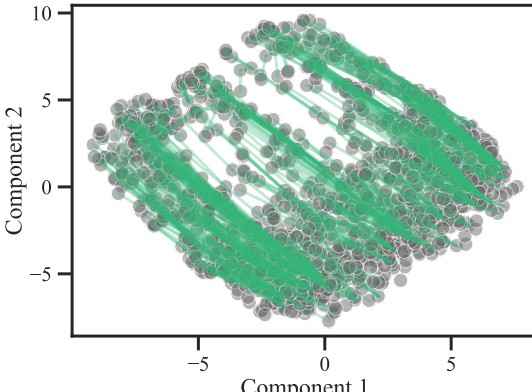

*Figure 21.* Parent-child token relations analyzed through principal component analysis of token embeddings from our medium tokenizer in a small model. Children and their first parents are connected.

## C.9. Learned motifs

Our tokenizer employs a vocabulary of frequent motifs to encode time series. To enhance interpretability, we illustrate selected patterns learned by our tokenizer with medium compression (see table 1). Note that the vocabulary also includes shifted and scaled variants of these motifs along the y-axis.

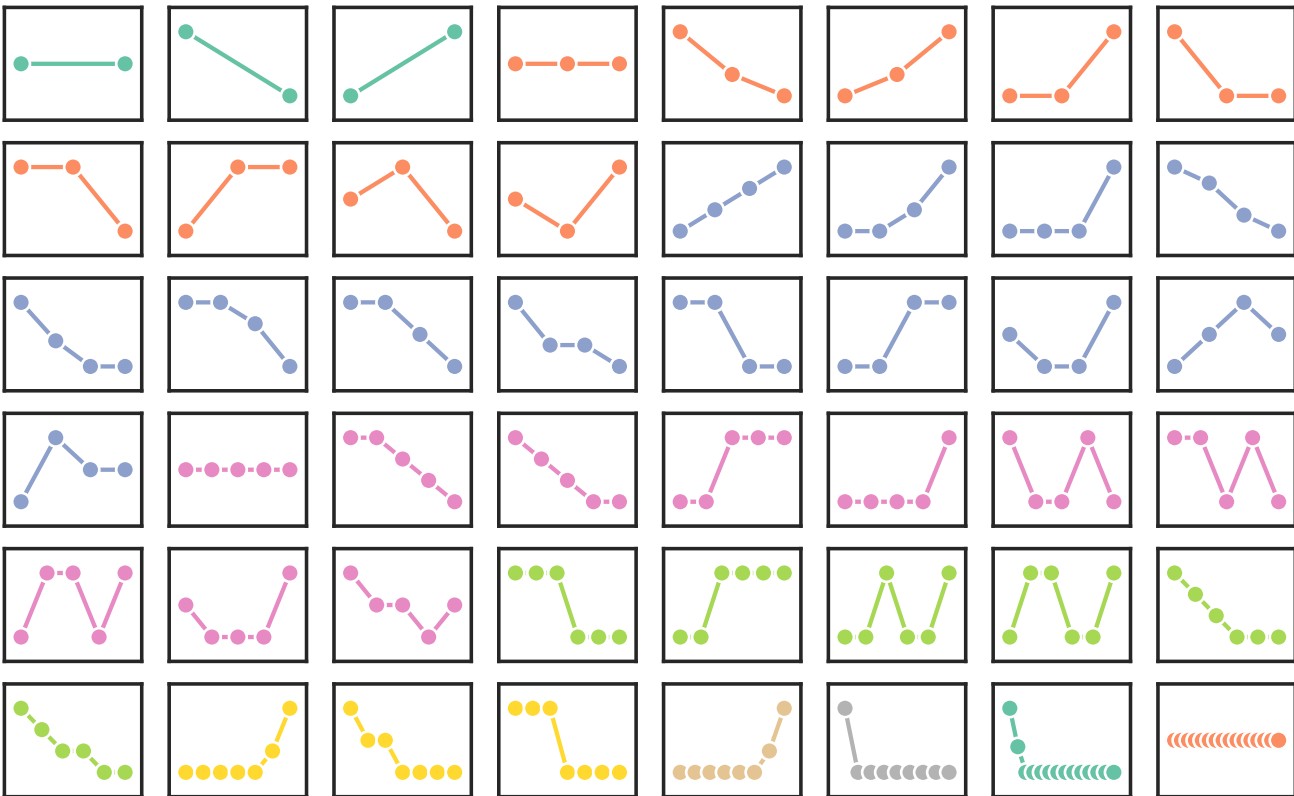

*Figure 22.* Visualization of motifs which our medium compression tokenizer uses to encode time series. Colors indicate motif length.

