# OpenReview forum: "Byte Pair Encoding for Efficient Time Series Forecasting"
_ICML.cc/2026/Conference — ICML 2026 regular_

### Official Review · Reviewer_PNV5 · 2026-03-10

**Soundness:** 3
**Presentation:** 4
**Significance:** 3
**Originality:** 3
**Overall Recommendation:** 4
**Confidence:** 3

**Summary:**

This paper proposes a BPE-style motif tokenizer for time series: it discretizes values into symbols, then learns a variable-length motif vocabulary via frequent pair merges to compress sequences. A simple conditional decoding step (closed-form conditional means) reduces discretization error without adding inference cost. Experiments on zero-shot forecasting across five datasets show improved MSE and better efficiency than sample-wise tokenization and basic patch variants.

**Compliance With Llm Reviewing Policy:**

Affirmed.

**Key Questions For Authors:**

1. Can you add an equal-raw-context or equal-compute comparison to disentangle compression benefits from longer effective history?

2. Have you tested stronger patch baselines (e.g., overlap/stride tuning) under the same “tokenization-only change” principle?

3. In strict zero-shot deployment, what data is used to fit the conditional decoding table, and how sensitive is it to distribution shift?

4. Please clarify how you avoid any potential overlap/leakage between tokenizer training data and evaluation datasets/sources.

**Limitations:**

Yes. The approach may be sensitive to discretization choices and to shifts in conditional statistics used for decoding; the study is also mainly limited to univariate forecasting.

**Strengths And Weaknesses:**

Strengths

1. Clear and modular contribution: motif-based tokenization + lightweight conditional decoding.

2. Empirical results are consistent across multiple datasets/model sizes, with meaningful speed–quality gains.

3. The conditional decoding idea is simple, practical, and potentially reusable.

Weaknesses

1. Fixed-token comparisons also change the raw context length per model input, which may confound where gains come from.

2. Patch baselines may not reflect stronger patching variants (e.g., overlap/stride/embeddings).

3. Conditional decoding relies on estimated conditional statistics; robustness under domain shift is not fully characterized.

---

> ### Author Rebuttal · Authors · 2026-03-29
>
> Dear Reviewer PNV5,
>
> Thank you for taking the time to read our paper and for your valuable suggestions. We have conducted new experiments addressing your comments (https://figshare.com/s/fedd9b3c5996babf47a6). We are happy to answer your questions below.
>
> **W1/Q1:** Effective context length \
> **A:** We originally compared tokenization methods using a fixed input token number. This way, the transformer encoder utilizes an equal compute-budget and efficiency gains are solely due to compressed token generation in the decoder. As you pointed out, this leads to different effective input token length. For this rebuttal, we performed additional experiments utilizing the same effective context length of 384 time series samples for all tokenization methods. Specifically, we retrained Chronos baselines with 384 input samples (patches already use this length). Our results (new tab. 6) show that motif-based tokenization is still superior to Chronos baselines. Additionally, Chronos baselines are now slower due to longer input context.
>
> **W2/Q2:** Patches with stride \
> **A:** Thank you for pointing us to that. We now include new patch-based Chronos baselines with half-overlapping patches. Motif-based tokenization also outperforms these new baseline models (new tab. 2). This highlights the performance of our tokenizer. \
> Additionally, we now compare to in total 4 patch-based literature models. Here, our tokenizer is superior in 55 out of 63 cases (new tab. 3).
>
>
> **W3/Q3:** Conditional decoding distribution \
> **A:** For strict zero shot deployment, we quantize time series from the Chronos dataset (our training dataset) into discrete symbols. In a second step, we dequantize these quantized time series and analytically compute conditional decoding distributions to minimize the error between the dequantized and the original sequence. In this setting no model is involved and only the Chronos pretraining dataset is used (lines 992-994). \
> In our pure zero shot evaluation on 5 datasets, conditional decoding mitigates 31.9% of the quantization error (line 284, right; tab. 14). For this rebuttal, we perform additional evaluations on the Fev-benchmark, which consists of 7 domains and is curated from 96 datasets. Even in this very diverse setting, pure zero shot conditional decoding mitigates 29.6% of the quantization error on average. This demonstrates the robustness and zero shot generalization of conditional decoding distributions.
>
>
> **Q4:** Data leakage \
> **A:** We train our tokenizers and our foundation models strictly on the Chronos pretraining dataset. We evaluate tokenizers and models on the (validation or test splits) of the ETTh1, ETTm1, Weather, Electricity, and Traffic datasets. These evaluation datasets are not included in the Chronos pretraining dataset (see [1], page 31). \
> [1] Ansari et al. Chronos: Learning the language of time series. In Transactions on Machine Learning Research, 2024.
>
> **L:** Multivariate time series \
> **A:** We agree that multivariate time series are of practical interest.  To apply our tokenization to multivariate time series we would follow current literature that patches variates individually [2,3,4,5]. While, the design of the Chronos model does not make it compatible with multivariate forecasting, applying byte-pair encoding on the time-axis for every individual varite would be directly compatible with recent foundation models in the multivariate literature [2]. \
> [2] Woo et al. Unified training of universal time series forecasting transformers. arXiv:2402.02592, 2024.
> [3] Gao et al. Units: A unified multi-task time series model. In Advances in Neural Information Processing Systems, 2024.
> [4] Cohen et al. Toto: Time series optimized transformer for observability. arXiv:2407.07874, 2024.
> [5] Ekambaram et al. Tiny time mixers (ttms): Fast pre-trained models for enhanced zero/few shot forecasting of multivariate time series. In Advances in Neural Information Processing Systems, 2024.
>
> **Action:** Thank you for your valuable suggestions. We will include our new results and discussions in our manuscript. We believe this further strengthens our contribution. We hope we could resolve your concerns and kindly ask you to reconsider your score.

---

### Official Review · Reviewer_3zmC · 2026-03-11

**Soundness:** 3
**Presentation:** 3
**Significance:** 3
**Originality:** 3
**Overall Recommendation:** 4
**Confidence:** 3

**Summary:**

This paper proposes a motif-based tokenization scheme for time series forecasting inspired by byte pair encoding. The method first quantizes samples into discrete symbols, then recursively merges frequent adjacent symbols into a vocabulary of motifs of variable length, so that repetitive or simple patterns can be represented by fewer tokens. The paper also introduces conditional decoding, a lightweight post-hoc dequantization method based on first-order symbol transitions, and evaluates the approach in zero-shot forecasting settings.

**Compliance With Llm Reviewing Policy:**

Affirmed.

**Final Justification:**

I keep my original score.

**Key Questions For Authors:**

See weaknesses.

**Limitations:**

yes

**Strengths And Weaknesses:**

Strengths:

1. The core idea is intuitive and reasonably well motivated. Using BPE-style merges over quantized symbols is a sensible adaptation of a successful compression/tokenization principle to time series. I also appreciated that the method is not only pitched as compression, but as a representation change that could alter the learning problem in useful ways.
2. The presentation of the high-level pipeline is strong.
3. The paper includes useful qualitative analyses instead of stopping at benchmark tables.

Weaknesses:

1.The zero-shot story becomes blurry once conditional decoding is introduced, and this matters a lot for the claimed gains. If part of the reported improvement comes from fitting a target-dataset-specific post-hoc decoder, then the paper should separate “pure zero-shot” from “zero-shot model + dataset-specific calibration.” Right now the presentation makes the conditional decoding gains look more directly comparable to zero-shot model comparisons than they really are.
2. On Page 5, the authors say the patch-based Chronos version “alter[s] only the tokenization method” but also “replace[s] the cross-entropy loss function with MSE.” That is already more than a tokenization change. The output space, objective, and training signal all change. As a result, the comparison in Table 3 cannot be interpreted purely as “motifs vs patches.”
3. On Page 3, the paper states that the algorithm scales linearly with sequence length, (O(n)). That may be true for a particular implementation of applying a fixed vocabulary to a sequence, but the paper does not provide enough detail to justify the claim for either vocabulary construction or tokenization with hierarchical merges.

---

> ### Author Rebuttal · Authors · 2026-03-29
>
> Dear Reviewer 3zmC,
>
> Thank you for taking the time to read our paper and for your valuable suggestions. We have already revised our paper based on your comments and are happy to answer your questions below.
>
> **W1:** Conditional decoding in zero shot and data-dependent setting \
> **A:** Thank you for pointing us to that. We are sorry for the confusion. We already explore pure zero-shot conditional decoding in appendix C3 and data-dependent conditional decoding in sec. 5.3. We tried to distinguish both by naming them "data- and model-independent" and "data- and model-dependent". We have already revised our paper to introduce conditional decoding more clearly: \
> *"Conditional decoding has two major uses: a) It can be applied in pure zero shot fashion where it is data- and model-independent (appendix C3). Here, conditional decoding can only mitigate the quantization error. b) Conditional decoding can be used as a fast and simple domain adaptation method where only the global optimum for its few parameters (484) is analytically computed. This domain adaptation / dataset calibration takes a matter of seconds on a single CPU core compared to retraining a whole foundation model and requires only little data (sec. 5.3). Here, conditional decoding is model- and data-dependent."* \
> To further distinguish results, we will denote results as $MSE^{cd,zs}$ for zero-shot conditional decoding and $MSE^{cd,da}$ for domain adaptation conditional decoding.
>
>
> **W2:** Comparison of motif-based tokenization and patches \
> **A:** Thank you for mentioning this detail. Motifs are discrete in nature.
> This is a deliberate property of our tokenization, as it transforms a regression problem in continuous space into a classification problem. Due to the max decision involved when selecting a motif as the prediction, MSE loss is infeasible for discrete motifs (since the max operation stops gradient propagation), and we therefore use cross‑entropy loss. /
> Conversely, prior work embeds patches into continuous space and regresses the prediction. Here, cross entropy loss is infeasible and MSE is commonly used. \
> This results in a straightforward design decision: one could either use motif-based tokenization together with cross entropy loss or patches together with MSE loss. Other permutations are infeasible. We therefore believe the comparison of motifs and patches is fair as discrete motif space or continuous patch space determines the loss function. Put differently, one might see the cross entropy loss as a feature enabled by motif-based tokenization. \
> Besides patches, we compare motif-based tokenization to single-sample tokenization in Chronos models. This is the only possible case where cross entropy loss is common and tokenization the single difference. \
> Based on your valuable comment we will clarify this important detail in our manuscript.
>
> **W3:** Complexity of motif-based tokenizaton \
> **A:** Thanks for bringing this up. We agree adding this missing detail improves our work. Applying a fixed-sized vocabulary of size $|\Psi|$ to a time series of length $n$ has a complexity of $O(|\Psi| \cdot n)$. In this naive / worst case implementation, the whole time series is compared with each merge entry in the vocabulary (eq. 4 and algorithm 2 on page 11). In practice (after the tokenizer training itself) the vocabulary has a fixed size, $|\Psi|$ is a constant factor and we abbreviated the complexity with $O(n)$ being linear in sequence length. \
> Besides that, tokenization has irrelevant computation overhead in our experiments being <0.5% in run time of our fastest models (line 646).
>
> **Action:** We already included this additional information, discussions, and clarifications into our revised manuscript. Again, thank you for pointing us to these missing details, we believe this improved our work. \
> For this rebuttal, we conducted further experiments comparing against new baseline and literature models. We also include additional datasets. In these experiments, motif-based tokenization further outperforms single-sample tokenization and patches (see https://figshare.com/s/5d1cf236f9fc327c901a).
> We hope we could resolve your concerns and kindly ask you to reconsider your score.

---

> > ### Author Rebuttal · Reviewer_3zmC · 2026-04-02
> >
> > Fully resolved - My concerns have been adequately addressed.

---

### Official Review · Reviewer_hNL2 · 2026-03-11

**Soundness:** 3
**Presentation:** 3
**Significance:** 4
**Originality:** 3
**Overall Recommendation:** 5
**Confidence:** 3

**Summary:**

This paper proposes a BPE-inspired, pattern-centric tokenization method for time series forecasting. Rather than assigning each token to a fixed-length chunk of samples, it learns a discrete vocabulary of recurring motifs and represents the series using variable-length tokens, allowing adaptive compression of simple or repeated patterns. This can substantially reduce sequence length and improve the efficiency of transformer-based forecasting models. The paper also introduces conditional decoding, a lightweight post-hoc method that aims to reduce discretization error and improve forecast accuracy without gradient-based optimization or extra inference overhead. Experiments on recent time-series foundation models show better forecasting performance together with large efficiency gains, and the authors further analyze adaptiveness, generalization, and interpretability of the learned token representations.

**Compliance With Llm Reviewing Policy:**

Affirmed.

**Key Questions For Authors:**

1. How well does the learned motif vocabulary transfer across domains or datasets? For example, if the tokenizer is learned on one domain and evaluated on another, how much do compression and forecasting performance degrade? Is retraining typically necessary, or does the learned vocabulary generalize robustly?
2. The paper claims that conditional decoding introduces no computational overhead. Could the authors clarify precisely what this means in practice, e.g., no additional model forward passes, negligible wall-clock overhead, or unchanged end-to-end inference latency?
3. How does the proposed method compare to other adaptive schemes, such as run-length-style encoding, learned segmentation, or event-based tokenization? This would help determine whether the improvements arise specifically from the BPE-style motif construction or more broadly from adaptive compression.
4. Is the benefit of the proposed tokenization specific to forecasting, or might it also extend to other time-series tasks such as classification, anomaly detection, or imputation? Even a brief discussion of applicability beyond forecasting would help clarify the broader impact of the method.

**Limitations:**

1. The method relies on a motif vocabulary learned from data. If the training corpus is not representative, or if the deployment domain differs substantially, the tokenizer may fail to compress well or may produce less useful tokens. This raises concerns about robustness under domain shift.
2. Although the paper appears to include some ablations, the method still introduces several design decisions: quantization/discretization granularity, vocabulary size, merge criteria, and token occurrence thresholds. These choices may affect both compression and forecasting accuracy, and may require retuning across datasets.
3. The approach gains efficiency by discretizing and merging patterns, but this can introduce reconstruction or representation error, especially for subtle local variations or rare events. Conditional decoding is intended to mitigate this, but it also indicates that the base tokenization may discard useful fine-grained information.

**Strengths And Weaknesses:**

S1. The paper addresses a real limitation of fixed-size time-series tokenization: simple or repetitive patterns can still produce long token sequences, and transformer cost scales strongly with token count. The motivation for adaptive compression is therefore compelling.
S2. The motif-based, BPE-inspired tokenization scheme is a concrete algorithmic idea rather than an incremental tweak. It is easy to understand at a high level, appears implementable in practice, and has direct implications for both efficiency and forecasting quality.
S3. Beyond reporting headline forecasting and efficiency results, the paper includes additional analyses on adaptiveness, generalization, and representation properties. These help support the broader technical narrative and make the work more convincing.

W1. Because the method relies on a learned motif vocabulary, its effectiveness may depend on how representative the training corpus is. If test-time patterns differ substantially from training-time motifs, both compression and predictive performance may degrade.
W2. The behavior of the method may be sensitive to choices such as vocabulary size, motif construction, and merge criteria. The paper would be stronger with a clearer ablation or robustness study showing stability across these design parameters.
W3. The claim of “no computational overhead” is appealing, but in practice even lightweight post-hoc refinement can introduce latency or implementation complexity. It would help to clarify whether this means asymptotically negligible overhead, no additional model forward passes, or truly unchanged wall-clock inference time.

---

> ### Author Rebuttal · Authors · 2026-03-29
>
> Dear Reviewer hNL2,
>
> Thank you for taking the time to read our paper and for your valuable comments. We are happy to answer your questions below.
>
> **Q1/L1** Vocabulary generalization \
> **A:** We train our tokenizer on the Chronos dataset and evaluate in zero shot setting covering new domains and datasets (ETTh1, ETTm1, Weather, Electricity, Traffic). Our main results (tab. 2,3) demonstrate that our motif vocabulary generalizes very well outperforming single-sample tokenization and patches. \
> We now conduct evaluations on the Solar dataset and the Fev-benchmark, which consists of 7 domains and is curated from 96 datasets. Here, our motif-based tokenization also outperforms single-sample tokenization and patches (https://figshare.com/s/5d1cf236f9fc327c901a, tab. 1,2,3,4). This demonstrates the generalization of our motif vocabulary to very diverse settings. \
> Besides these promising results, we agree that there might be domains where our vocabulary does not generalize. This might be a general limitation of foundation models and not just one of tokenization. In the worst case, our tokenizer falls back to single sample tokenization and we suggest simply training a new tokenizer, which only takes 3.8 hours on a single CPU core (line 654). For our work, however, retraining was never necessary.
>
> **Q2** No computational overhead for conditional decoding \
> **A:** Thank you for pointing us to this detail. With "normal" decoding, discretized symbols are detokenized to time series samples through a dictionary look-up with dictionary length $M$ (number of discretization bins). For conditional decoding, the same dictionary just has more entries ($M^2$). Dictionary look-ups are implemented with constant complexity O(1) through hash tables, independent of the number of entries. This is why conditional decoding has *no* additional overhead in theory and in our measurements (unchanged end-to-end latency).
>
> **Q3:** Other adaptive encoding schemes \
> **A:** This is an interesting idea. Motif-based tokenization simplifies the forecasting task from regressing every sample to selecting even complex patterns as building blocks from a vocabulary. In contrast, run-length encoding can not represent complex patterns in compressed form.
> Further, our motif-based tokenization is especially appealing as it embeds time series into discrete space. This allows future time series models to directly rely on advanced architectures from natural language processing, opening a new field of time series models. Additionally, motif-based tokenization compresses time series exponentially with vocabulary depth which is very efficient (line 1249). In contrast to event-based encoding, motif-based tokenization can also compress "dense" time series (fig. 5). Finally, our discrete motif embeddings learn distinct time series properties such as means and standard deviations, which can not be easily achieved with run-length encoding (sec. 5.7).  \
> While motif-based tokenization is a promising encoding method, future work might explore related encoding schemes.
>
>
> **Q4:** Other time series tasks \
> **A:** Thank you for pointing us to this. We strongly believe that motif-based tokenization should also be used for other tasks such as classification or anomaly detection. Here, certain clases or anomalies might be directly linked to a subset of motifs. Overall, we view motif-based tokenization as universal time series encoding method for foundation models.
>
> **L2:** Design decisions \
> **A:** Our approach introduces new hyperparameter as you pointed out. We ablate all these hyperparameters (sec. 5.5, 5.6). The strong performance in our zero shot evaluation indicates that these hyperparameters generalize well. Nonetheless, we would like to give further hands-on guidance on how to choose them: \
> - *quantization granularity:* $M=22$ bins is best in 15 out of 25 settings (line 378, left).
> - *token occurrence threshold:* $p_{min}=4000$ is a good balance (line 350, right).
> - *vocabulary size:* is determined automatically through $p_{min}$
> - *training dataset size:* larger is better but 100 k time series is already very good (sec. 5.6)
>
> In appendix C1, we further evaluate 500 different hyperparameter combinations and find these to be best.
>
> **L3:** Quantization discards information \
> **A:** You are correct that quantization discards information. Conditional decoding is designed to mitigate information loss and achieves up to 96% reconstruction (line 284, right). However, there might be settings where very subtle changes might be important. In these cases, we suggest choosing more fine-grained discretization intervals or placing them in regions of interest. In general, the good predictive performance of our models demonstrates the effectiveness of our tokenization approach, outperforming even continuous patches in zero shot generalization settings.
>
> **Action:** Thank you again for your valuable comments. We believe including these discussions further strengthens our work.

---

> > ### Author Rebuttal · Reviewer_hNL2 · 2026-03-31
> >
> > My concerns have been addressed.

---

### Official Review · Reviewer_EA7p · 2026-03-13

**Soundness:** 3
**Presentation:** 3
**Significance:** 3
**Originality:** 3
**Overall Recommendation:** 4
**Confidence:** 5

**Summary:**

The authors explore a well known NLP method of tokenisation "Byte Pair Encoding" and its application in Time Series Foundational Models(TSMF). To this end they quantise the time series into bins and then merge them iteratively with adjacent token getting clubbed into motifs of varying lengths. Additionally they introduce a lightweight post-hoc correction to decode the tokens to reduce quantisation error when mapping back from tokens to continuous values. They train Chronos backbone and show 36% improvement in MSE and 1990% improvement in efficiency.

**Compliance With Llm Reviewing Policy:**

Affirmed.

**Final Justification:**

My final recommendation is weak accept.

**Key Questions For Authors:**

In addition to the questions asked in the weakness. Additional questions are-

1- Can authors comment on the standard deviation reported in Table 9, what dataset properties lead to this? Any reasoning behind the high standard deviation of BPE vs Chronos?

2- Can authors provide comparison on Fev-bench, Gifteval ?

3- Can authors provide emperical evidence or theoretical proof behind the first-order Markov assumption? When does the assumption fail?

**Limitations:**

Yes

**Strengths And Weaknesses:**

**Strengths**

- **Lightweight Decoding** : The lightweight decoding is  elegant which is a a gradient-free, zero-overhead post-hoc correction that exploits the finite motif vocabulary and continuous nature of time series to reduce quantisation error.

- **Presentation**: The paper is clearly written and easy to follow, with well-designed algorithm pseudocode (Algorithms 1-2) and pipeline diagrams that make the two-stage tokenisation process easy to understand and follow.

-  **Ablations and Analysis** : The ablation studies are detailed and well studied against Chronos baselines, systematically varying compression, quantisation, token occurrence, with an insightful PCA analysis showing that learned motif embeddings meaningfully capture statistical moments, trends, and hierarchical BPE parent-child relationships.


**Weaknesses**

- **Novelty** : Though byte pair encoding in TSFM is novel though its application is limited to the vocabulary style TSFMs like Chronos and in table 3 the performance degradation in 3/5 datasets to MORAI demonstrates its limited usability to vocabulary style TSFMs.

- **Comparison with baselines**: Important baselines both TSFM[1][2] based and native to time series[3][4][5] forecasting ones are missed. Furthermore LightGTS (ICML 2025) which delves into similar idea of  periodical tokenization that adaptively splits patches based on the intrinsic period of the dataset is missed.

- **Limited Datasets** : Given the TSFMs are usually evaluated on bigger benchmarks like Fev-bench and  Gift Eval, the current evaluation set is very limited. Furthermore the results on the evaluated benchmarks are mixed.

-  **Over exaggerated efficiency claim** : The claim of 1990% efficiency is over exaggerated as it is compared with the biggest Chronos model a fair comparison would be with a similar-sized Chronos variant which is present in Table 12 in the appendix.

[1] Shi et al., "Time-MoE: Billion-Scale Time Series Foundation Models with Mixture of Experts", ICLR 2025
[2] Das et al., "A Decoder-Only Foundation Model for Time-Series Forecasting", ICML 2024
[3] Nie et al., "A Time Series is Worth 64 Words: Long-term Forecasting with Transformers (PatchTST)", ICLR 2023
[4] Liu et al., "iTransformer: Inverted Transformers Are Effective for Time Series Forecasting", ICLR 2024
[5] Eldele et al., "TSLANet: Rethinking Transformers for Time Series Representation Learning", ICML 2024
[6] Liang et al., "LightGTS: Light Guided Time Series Tokenization", ICML 2025

---

> ### Author Rebuttal · Authors · 2026-03-29
>
> Dear Reviewer EA7p,
>
> Thank you for taking the time to read our paper and for your valuable comments. We have conducted several new experiments (https://figshare.com/s/5d1cf236f9fc327c901a) and have revised our paper. We are happy to answer your questions.
>
> **W1:** Novelty, Limited application, and effectiveness \
> **A:** We agree that novelty, applicability, and effectiveness are central:
>
> **Novelty:** Beyond motif-based tokenization, we introduce conditional decoding as a novel lightweight post-hoc optimization that applies broadly to any method forecasting in discrete space over continuous data.
>
> **Applicability/Effectiveness**: While numerous works focus on continuous architectures, we believe discrete time series foundation models are an underexplored and promising direction. In a controlled comparison where tokenization is the only variable, discrete, motif-based tokenization outperforms all continuous, patch-based Chronos baselines, improving MSE by 21.3% on average (paper tab. 3). Broader comparisons to literature models are confounded by different backbones and proprietary training data. Nevertheless, new experiments against Time-MoE and LightGTS (new tab. 3) further demonstrate superior performance against state-of-the-art foundation models.
>
> **W2:** Comparison with baseline \
> **A:** We compare our tokenization to patches and single sample tokenization in an atomic setting where we isolate tokenization from effects of different backbone architectures and training datasets.
> We now add two new patch-based Chronos baselines with half-overlapping patches. Here, our tokenizer is superior on all datasets (new tab. 2). \
> However, we see the importance to also compare more broadly to literature. For this rebuttal, we evaluate against Tome-MoE [1] and LightGTS [6]. In our new results, motif-based tokenization outperforms Tome-MoE in all cases and is superior to LightGTS in 5 out of 7 cases (new tab. 3). This demonstrates the advantage of motif-based encoding. \
> We do not compare to [2] due to the lack of public training code and the use of proprietary data. Moreover, [3][4][5] are not foundation models and are not designed or trained for zero-shot inference, which is the focus of our work.
>
> **Q2/W3:** Fev-bench \
> **A:** Thank you for pointing us to that. We agree that further evaluation on broad benchmarks will strengthen our contribution. We conduct new evaluations on the Fev-benchmark (which includes GIFT‑Eval) and the Solar dataset. On both datasets motif-based tokenization outperforms single-sample tokenization and patches (new tab. 1,2,3,4).
>
> **W4:** Efficiency claim \
> **A:** Our tokenization improves both efficiency and MSE. This enables us to choose smaller models while maintaining predictive quality due to our tokenizer. Coming from a practitioner's perspective, we reported overall efficiency gains including the ones due to smaller models as this is enabled by our tokenizer in the first place. (paper l. 240-244, right) \
> However, we agree that this might be confusing. We have revised our paper and now report both: *"Motif-based tokenization boosts efficiency by 196% on average while simultaneously improving MSE by 36%. This enables us to additionally choose smaller models while maintaining predictive performance, resulting in greater overall speed up of 1990% for practical applications."*
>
> **Q1:** Standard deviation in Table 9 \
> **A:** This is an interesting aspect. Standard deviations are measured in zero shot setting. This introduces some noise. Most notably, the original Chronos model fails to generalize to the Traffic dataset, resulting in a high standard deviation (paper tab. 9,2). \
> Computing the average standard deviation among all datasets in tab. 9 reveals that our motif-based tokenization has the lowest standard deviation (0.042) followed by patches (0.047) and the original chronos model (0.059).
>
> **Q3:** First-order Markov assumption \
> **A:** Conditional decoding translates tokens back to time series representations conditioned on the previous token. By design, this follows the first-order Markov assumption and has $M^2$ parameters, where $M$ is the number of quantization bins. \
> While longer look-back windows, e.g, conditioning on the 2 or 3 previous tokens, are possible, these have exponentially more parameters ($M^3$, $M^4$). These higher-order Markov assumptions might lead to better performance in special cases but might also hinder generalization or overfit. In our new experiment (new tab. 5), higher-order conditional decoding only marginally improves predictive performance while having substantially more parameters. This demonstrates the effectiveness of our first-order method.
>
> **Actions:** Based on your valuable comments, we will include these new experiments and discussions in our paper. We have already revised our manuscript for more clarity (e.g., W4, Q1) and believe this improves our work. We hope we could resolve all your concerns and kindly ask you to reconsider your score.

---

> > ### Author Rebuttal · Reviewer_EA7p · 2026-04-02
> >
> > Thank you for the substantial rebuttal. W2 (baselines), W4 (efficiency), Q1 (std deviation), and Q3 (Markov ablation) are adequately addressed.
> > However, the following concerns remain:
> > 1. Novelty (W1) persists. The method remains an adaptation of BPE to vocabulary-style TSFMs and does not generalize to continuous-embedding architectures. Can the authors comment on extensibility beyond discrete vocabulary models?
> > 2. Performance gaps. The method still underperforms Moirai on Weather (0.236 vs 0.161), ETTh1 (0.459 vs 0.396), Traffic (0.574 vs 0.406), and LightGTS on Weather (0.236 vs 0.157) and ETTh1 (0.459 vs 0.391).
> > 3. GIFT-Eval not evaluated. Fev-benchmark is reported but GIFT-Eval has not been separately evaluated as requested.
> > 4. Full Fev-bench results. Only aggregated numbers are shown can the authors post per-dataset results across all 96 datasets / 7 domains?

---

> > > ### Author Response · Authors · 2026-04-03
> > >
> > > Dear Reviewer EA7p,
> > >
> > > Thank you for your reply and for detailing your questions. We would like to address them below and have conducted new experiments (https://figshare.com/s/a407e342c8d39ff88e3f).
> > >
> > > **Q1:** Applicability to time series architectures \
> > > **A:** This is an important aspect, which we address from three perspectives:
> > >
> > > In a new experiment, we demonstrate that we can refine the predictions of patch‑based models from the literature using our proposed motif‑based tokenization in a post‑hoc fashion. To this end, we first quantize the predictions of patch‑based models to reduce noise. In a second step, we transform them back into time‑series representations using conditional decoding. Our results (new tab. 1) show that we can improve the forecasting quality of patch‑based models by 5% on average, without modifying or retraining the model itself. This post‑hoc application makes our tokenization directly compatible with current continuous‑embedding models in the literature.
> > >
> > > Moreover, we successfully demonstrate in our paper that the discrete Chronos model can be easily transformed into a continuous patch-embedding architecture by only altering the tokenization and loss function. The reverse is also possible: one could take a continuous-embedding architecture, replace patches with our motif-based tokenization, and swap the MSE loss for a cross-entropy loss. In this way, our tokenization is applicable to current time series architectures with only minor modifications.
> > >
> > > Beyond this immediate applicability, we believe motif-based tokenization could become increasingly important as the field evolves. With motif-based tokenization, we propose a promising discrete alternative that outperforms continuous patches in direct comparison. Moreover, these discrete models can now also adopt advances from other domains, such as natural language processing.
> > >
> > > **Action:** We demonstrate a direct application of motif-based encoding as a post-hoc refinement to improve predictions of existing patch-based models. We further discuss how existing continuous architectures could be adopted too make use of Byte-Pair Encoding beyond our post-hoc refinement.
> > >
> > > **Q2:** Performance differences to patch-based models \
> > > **A:** We perform an atomic comparison of patches and motif-based tokenization, isolating tokenization from the effects of different backbone architectures and pretraining datasets. Here, motif-based tokenization outperforms patches on 6 out of 7 datasets (last rebuttal tab. 2).
> > >
> > > In our comparison to literature models, motif-based tokenization outperforms *all* MOMENT models, *all* Time-MoE models, LightGTS in 5 out of 7 cases, and Moirai in 4 out of 7 cases (please note that Moirai is trained on the Traffic dataset, whereas we perform zero-shot generalization, making this comparison confounded). Among datasets, the average MSE of motif-based tokenization (0.427) is substantially better than the one of Moirai (0.490) and LightGTS (0.534). In total, motif-based tokenization outperforms literature patch-based models in 55 out of 63 cases (last rebuttal tab. 3). While it is expected that no *single* method is superior in *every* setting — especially when the pretraining dataset and backbone architecture change — these results demonstrate the strong performance of motif-based tokenization compared to patches.
> > >
> > > **Q3:** FEV and GIFT-Eval \
> > > **A:** Thank you for pointing this out. We evaluated on FEV because FEV and GIFT-Eval have substantial overlap. Specifically, FEV is sourced from GIFT-Eval and additional datasets and therefore already contains GIFT-Eval. As stated by Shchur et al. (2026): “We  source time series datasets from established collections including the Monash repository (Godahewa et al., 2021), GIFT-Eval (Aksu et al., 2024), [...]”
> > >
> > > Aksu et al., GIFT-Eval: A Benchmark for General Time Series Forecasting Model Evaluation, 2024.
> > > Godahewa et al., Monash Time Series Forecasting Archive, 2021.
> > > Shchur et al., fev-bench: A Realistic Benchmark for Time Series Forecasting, 2026.
> > >
> > > **Q4:** Aggregated FEV results \
> > > **A:** We presented aggregated FEV results, i.e., a single MSE for the whole benchmark, following the FEV authors (Shchur et al., 2026) and for clarity. We now also show individual results per dataset (new tab. 2, page 2). Please note that we exclude datasets with non-nummeric values (NaN, Inf). Some of the segments were also shorter than our input length plus prediction length.
> > >
> > > Thank you for discussing these important details with us. We hope we were able to resolve your concerns.

---

### Decision · Program_Chairs · 2026-04-30

**Decision:**

Accept (regular)

**Comment:**

This paper introduces a practical, BPE-inspired tokenization scheme and conditional decoding method for time series forecasting. Reviewers unanimously praised how the approach adaptively compresses repetitive temporal patterns into variable-length motifs, effectively resolving the computational bottlenecks of fixed-length tokenization with zero decoding overhead. During the rebuttal, the authors successfully addressed initial concerns by providing comprehensive Fev-benchmark evaluations and direct comparisons against state-of-the-art models like Time-MoE and LightGTS. The authors should include the above experimental results and discussions into the final version. In addition, the AC suggests to consider another benchmark TSFM-Bench (KDD 2025).

Given the above, assuming that the authors will include the additonal discussions and experiments in the rebuttal into the final verison, I recommend this paper for Acceptance.